

# Attribution of climate change and human activities to streamflow variations with a posterior distribution of hydrological simulations

Xiongpeng Tang [1, 2, 3], Guobin Fu [4], Silong Zhang [1], Chao Gao [1], Guoqing Wang [2, 3], Zhenxin Bao [2, 3], Yanli Liu [2, 3], Cuishan Liu [2, 3], Junliang Jin [2, 3]

[1]Beijing Normal University at Zhuhai, Zhuhai 519087, China

[2]State Key Laboratory of Hydrology-Water Resources and Hydraulic Engineering, Nanjing 210098, China

[3]Nanjing Hydraulic Research Institute, Nanjing 210029, China

[4] CSIRO, Land and Water, Private Bag 5, Wembley, WA 6913, Australia

*Correspondence to*: Guoqing Wang (gqwang@nhri.com)

**Abstract:** Hydrological simulations are a main method of quantifying the contribution rate (CR) of climate change (CC) and human activities (HAs) to watershed streamflow changes. However, the uncertainty of hydrological simulations is rarely considered in current research. To fill this research gap, based on the Soil and Water Assessment Tool (SWAT) model, in this study, we propose a new framework to quantify the contribution rate of climate change and human activities based on the posterior histogram distribution of hydrological simulations. In our new quantitative framework, the uncertainty of hydrological simulations is first considered to avoid the phenomenon of "equifinality for different parameters", which is common in hydrological simulations. The Lancang River (LR) Basin in China, which has been greatly affected by human activities in the past two decades, is then selected as the study area. The global gridded monthly sectoral water use data set (GMSWU), coupled with the dead capacity data of the large reservoirs within the LR basin and the Budyko hypothesis framework, are used to compare the calculation result of the novel framework. The results show that (1) the annual streamflow at Yunjinghong station in the Lancang River Basin changed abruptly in 2005, which was mainly due to the construction of the Xiaowan hydropower station that started in October 2004. The annual streamflow and annual mean temperature time series from 1961 to 2015 in the LR Basin showed a significant decreasing and increasing trend at the $\alpha = 0.01$ significance level, respectively. The annual precipitation showed an insignificant decreasing trend. (2) The results of quantitative analysis using the new framework showed that the reason for the decrease in the streamflow at Yunjinghong station was 42.6% due to climate change, and the remaining 57.4% was due to human activities, such as the construction of hydropower stations within the study area. (3) The comparison with the other two methods showed that the contribution rate of climate change calculated by the Budyko framework and the GMSWU data were 37.2% and 42.5%, respectively, and the errors of the calculations of the new framework proposed in this study were within 5%. Therefore, the newly proposed framework, which considers the uncertainty of hydrological simulations, can accurately quantify the contribution rate of climate change and human activities to streamflow changes. (4) The quantitative results calculated by using the simulation results with the largest Nash-Sutcliffe efficiency coefficient (NSE) indicated that climate change was the dominant factor for streamflow reduction, which was in opposition to the calculation results of our new framework. In other words, our novel framework could effectively solve the calculation errors caused by the "equifinality for different parameters" of hydrological simulations. (5) The results of this case study also showed that the reduction in the streamflow in June and November was mainly caused by decreased precipitation and increased evapotranspiration, while the changes in the streamflow in other months were mainly due to human activities such as the regulation of the constructed reservoirs. In general, the novel quantitative framework



that considers the uncertainty of hydrological simulations presented in this study has validated an efficient alternative for quantifying the contribution rate of climate change and human activities to streamflow changes.

**Key words:** Attribution analysis; Climate change; Human activities; Lancang River Basin; Uncertainty; SWAT model

## 1. Introduction

Both the hydrological cycle of the watershed and water resource systems are deeply influenced by climate change (CC) and human activities (HAs) (Bao et al., 2012; Chandesris et al., 2019; Han et al., 2019; Teuling et al., 2019). Climate change mainly refers to changes in precipitation and evapotranspiration that are caused by rising temperatures and water vapor (Hegerl et al., 2015), while the impact of human activities is mainly reflected in the following aspects: reservoir construction changes the spatial and temporal distribution of streamflow processes (Hennig et al., 2013; Chandesris et al., 2019); land use changes change the

characteristics of the underlying surface of the watershed, in turn affecting the streamflow of the watershed (Yang et al., 2017); population increase leads to an increase in the amount of water used for domestic consumption (Teuling et al., 2019), etc. However, identifying which climate change and human activities are the main factors driving the changes in the water cycle of river basins is of great significance for water resource managers to formulate policies for sustainable water resource utilization (Dey and Mishra, 2017; Liu et al., 2017). If climate change is the dominant driving factor, then hydro-meteorologists need to assess the

future trends of meteorological factors, such as precipitation and temperature, to change their water resource management strategies in a timely manner. Conversely, if human activities are the dominant factor, water resource managers should evaluate whether the impact of these human activities exceeds the local water resource carrying capacity and then adjust their related policies (Fu et al., 2004).

Numerous published articles have focused on how to quantify the contribution rate (CR) of climate change and human activities

to the streamflow change in river basins (Bao et al., 2012; Kong et al., 2016; Chandesris et al., 2019; Han et al., 2019; Liu et al., 2019; Xie et al., 2019). In general, the commonly used methods of attribution analysis can be divided into the following three categories: 1) hydrological sensitivity and climate elasticity methods, such as the Budyko framework (Li et al., 2007; Liu et al., 2017); 2) hydrological simulation combined with scenario assumption methods (Liu et al., 2019); and 3) hydrological simulation prediction and separation methods (Han et al., 2019). What these three methods have in common is that they all need to first test

the annual streamflow sequence through non-stationary testing methods (such as the Mann-Kendall test, Moving t test and LePage test, etc.), and then divide the study period into the natural period (before the break point) and the impacted period (after the break point). The first type of method needs to first calculate the sensitivity of the basin's precipitation and potential evapotranspiration to hydrological variables, and then the hydrological changes caused by climate change can be calculated combined with the hydrological sensitivity parameters through the changes in precipitation and potential evapotranspiration in the impacted period

and natural period so that the CR of human activities is obtained based on the water balance equation (Li et al., 2007). The second type of method simulates multiple scenarios by changing one impact factor with other fixed factors to evaluate the contribution rate of the changed factor using lumped or distributed hydrological models (Liu et al., 2019). The core of these methods is the modelling of two situations where only one impact factor state has been changed, and the difference between the two simulation results is regarded as the influence of the changed factor. The third type of method mainly assumes that the streamflow process in

the natural period is not affected by human activities, and then uses the meteorological and hydrological data in the natural period to calibrate the parameters of the hydrological model to further predict the streamflow in the impacted period. Then, the impact





of climate change on streamflow can be calculated by subtracting the simulated natural period streamflow from that simulated in the impacted period (Bao et al., 2012). Among the three types of methods, the third method is the most widely used because it has the following advantages: 1) relatively small data requirements (one only needs to input the meteorological and hydrological data to the hydrological model); 2) relatively simple theoretical assumptions; and 3) can quantify the CR of CC and HAs to streamflow changes at the monthly scale.

Various related published articles are briefly reviewed as follows. Bao et al. (2012) used the variable infiltration capacity (VIC) model to investigate the impacts of CC and HAs on streamflow changes in the Haihe River Basin, China, and they concluded that human activity accounted for more than 70% of the decrease in streamflow at Guantai station. Wang et al. (2013) used a two-parameter hydrological model to quantify the contribution of climate change and human activities to streamflow changes in three river basins (i.e., Zhanghe, Chaohe, and Hutuo River), and they found that human activities were the dominant factor in streamflow changes. The above literature review shows that these studies all used hydrological simulations with fixed parameter sets to quantify the impact of climate change and human activities. As pointed out by Abbaspour et al. (2004) and Zhao et al. (2018b), there is a phenomenon of "equifinality for different parameters" (Beven, 2006) in hydrological calibration and simulation, which also means that we cannot ignore the uncertainty of model parameters in the process of quantifying the CR of CC and HAs to streamflow changes because two sets of parameters with the same performance (with the same Nash-Sutcliffe efficiency coefficient) may lead to very different results; this will further influence the decision-making of water resource managers to make effective and sustainable water resource utilization policies. In the last few decades, great progress has been made in evaluating the uncertainty of hydrological simulations (Beven and Binley, 1992; Abbaspour et al., 2004; Yang et al., 2008; Zhao et al., 2018b; Farsi and Mahjouri, 2019); however, in studies related to quantifying the CR of CC and HAs for streamflow changes, few studies have considered the uncertainty of hydrological simulations (Farsi and Mahjouri, 2019). According to our literature search, Farsi and Mahjouri (2019) first considered the uncertainty of hydrological simulations in the process of quantifying the CR of CC and HAs to streamflow changes, but they only constructed the posterior distribution of the contribution rates of climate change and human activities; in their research, they did not specify how to calculate the contribution rates of CC and HAs while considering the uncertainty of hydrological simulations. Therefore, to fill this research gap, in this study we propose a new method to quantify the contribution of climate change and human activities to streamflow changes considering the uncertainty of hydrological simulations, which in summary, is developed using the posterior histogram distribution of hydrological simulations.

The Lancang River (LR) is located in southwest China, and is the largest transboundary river in the Indo-China Peninsula; it is usually called the Mekong River (MR) after flowing out of China (Grumbine and Xu, 2011). The abundant water and ecological diversity of the Lancang-Mekong River Basin nurtures tens of millions of people in many countries along the Lancang-Mekong River. The upstream flow of the river provides guarantees for irrigation and fishery water use in the countries along the MR during the dry season, and the water conservancy facilities of the LR during the peak of the flood period also provide important engineering guarantees for downstream flood control (Piman et al., 2012; Piman et al., 2016). In the past three decades, a series of hydropower stations have been constructed in the LR Basin to meet the flood control and drought relief requirements of downstream countries and the power needs of Southwest China. Therefore, it is particularly important to quantify the contribution rate of climate change and human activities to streamflow changes in the LR Basin. However, so far, there are still few corresponding studies. Han et al. (2019) chose the Lancang River Basin (LRB) as the study area and then divided the research period into three periods, the natural period, transition period, and impacted period, and combined them with the construction time of six large hydropower stations in the LR area. Finally, they found that the contribution rate of human activities during the impact





period exceeded 95%, using the coupled routing and excess storage (CREST) model, which was probably due to the construction of the Nuozhadu hydropower station. However, there are still areas for improvement in their research: 1) the results of the hydrological simulation were relatively poor (with monthly NSE = 0.57 for the whole study period), and 2) the uncertainty involved in hydrological simulations was not considered.

In this paper, the breakpoint of the change in flow regimes was identified using the Mann-Kendall break point test. Then, the
study period was divided into a natural period (before the breakpoint) and an impacted period (after the breakpoint). The Soil and Water Assessment Tool (SWAT) model was used for monthly streamflow simulation at the Yunjinghong station. Next, the monthly SWAT model was calibrated and validated using the sequential uncertainty fitting procedure version 2 (SUFI-2) (Abbaspour et al., 2004). Uncertainty analysis was also conducted with the SUFI-2 method, and then the posterior histogram frequency distribution (HFD) of the CR of CC and HAs was obtained. Finally, the proposed quantification framework was
compared with two other methods: one was the Budyko framework, and the other was to use the LR Basin's gridded monthly sectoral water withdrawals in the period from 1971 to 2010 (Huang et al., 2018) together with the dead reservoir storage capacity data of the six constructed hydropower stations along the main stream of LR, to separate the contribution rate of human activities.

## 2. Study area and data sets

### 2.1 Study area

The Lancang River (LR) originates in the northeastern Tanggula Mountains, Qinghai Province, China, and flows through China's Qinghai Province, Tibet Autonomous Region and Yunnan Province. It is the largest international river in Southeast Asia, and it is called the Mekong River after it flows out of China. Its main stream has a total length of ~2161 km and a total catchment area of ~160000 km$^2$ (Li et al., 2017a; Han et al., 2019). The topography of the LR is characterized by high northern and low southern portions; the maximum elevation in the northern mountainous area can reach ~5871 meters, while the lowest elevation
in the downstream area is only ~547 meters (Fig. 1). This steep terrain difference also leads to the LR having a large potential for hydropower resources. During the past few decades, Huaneng Lancangjiang Hydropower Co., Ltd. constructed six large hydropower stations (i.e., Gongguoqiao, Xiaowan, Manwan, Dachaoshan, Nuozhadu and Jinghong) on the main stream of the LR to meet the demands for power and irrigation water in Southwest China (Fig. 1 and Table 1) (Xue et al., 2011; Hennig et al., 2013; Han et al., 2019). At the same time, the construction of these hydropower stations has greatly reduced the risk of flooding in
downstream countries and brought great convenience to using water for downstream agricultural irrigation. Detailed information on the six constructed hydropower stations is outlined in Table 1. These data are mainly collected from https://opendevelopmentmekong.net/topics/hydropower/, as well as from other published related literature (Xue et al., 2011; Hennig et al., 2013; Tilt and Gerkey, 2016; Han et al., 2019).

The LR features an arid climate in the upper mountainous areas, while the lower reaches are dominated by humid climates. The
average annual precipitation of the whole basin is ~ 870 mm based on a 55-year record (from 1961 to 2015) using the China Gauge-based Daily Precipitation Analysis (CGDPA) (Xie et al., 2007; Tang et al., 2019). Due to the influence of the westerlies and the Indian Ocean monsoon, the precipitation in the LR has obvious seasonal changes, and the precipitation from June to September accounts for more than 70% of the annual precipitation (Jacobs, 2002). Correspondingly, the streamflow of the LR also shows seasonality, and the floods are mostly concentrated from June to September.


**Fig. 1.** Locations of the Lancang River (LR) Basin, Yunjinghong hydrological station, constructed dams on the main stream of

the LRB, and main rivers and elevations (m).



**Table 1** Basic information for the six large constructed dams on the mainstream of the Lancang River

| Hydropower station | Manwan | Dachaoshan | Jinghong | Xiaowan | Gongguoqiao | Nuozhadu |
|---|---|---|---|---|---|---|
| Date of river closure | 1987.12 | 1997.11 | 2005.01 | 2004.10 | 2008.12 | 2007.11 |
| Date of complete construction | 1995.06 | 2003.10 | 2009.05 | 2010.08 | 2012.03 | 2014.06 |
| Drainage area ($10^4$ km$^2$) | 11.45 | 12.10 | 14.91 | 11.33 | 9.73 | 14.47 |
| Dead reservoir storage (km$^3$) | 0.668 | 0.371 | 0.81 | 4.35 | 0.316 | 10.3 |
| Total reservoir storage (km$^3$) | 0.92 | 0.94 | 1.40 | 15.3 | 0.365 | 22.7 |
| Installed capacity ($10^4$ kw) | 150 | 135 | 150 | 420 | 90 | 500 |

(Notation: Dead storage capacity refers to the storage capacity below the dead water level of the reservoir, which does not

participate in runoff regulation during the normal operation of the reservoir.)

**2.2 Data sets**

The China Gauge-based Daily Precipitation Analysis (CGDPA) product was developed by the China Meteorological

Administration (CMA) using data from ~2400 ground-based national weather stations across China (Xie et al., 2007; Shen et al., 2014; Tang et al., 2018). It provides daily precipitation, maximum temperature, minimum temperature, relative humidity, and wind speed data at a 0.25-degree spatial resolution from 1961 to 2015 (http://cdc.nmic.cn). Previous studies have successfully applied this product to multiple research areas in China (Tang et al., 2018; Han et al., 2019; Tang et al., 2019). The daily streamflow data from Yunjinghong station for the time period from 1961 to 2015 were collected from the Information Center of the Ministry

of Water Resources and the local water resources management department.

The digital elevation model (DEM) used in this study was downloaded from NASA's Shuttle Radar Topography Mission (SRTM) data bank at a spatial resolution of ~90 meters (http://srtm.csi.cgiar.org/), which was used to generate the watershed boundary, slope and sub-watershed data in the SWAT model (Arnold et al., 2012a). The Harmonized World Soil Database (version 1.2) (HWSD v1.2) at a spatial resolution of ~1 km was downloaded from the Food and Agriculture Organization of the United

Nations, and this data set contains two layers of soil. The land use and cover data with a spatial resolution of ~1km were collected from the Geospatial Data Cloud (http://www.gscloud.cn/). In this study, to analyze the land use change in the LR during the historical period, we collected five periods of land use data in the 1980s, 1990s, 2000s, and from 2010 to 2015.

The global gridded monthly sectoral water use (GMSWU) data set for 1971-2010 was obtained from https://zenodo.org/record/1209296#.XsJmiTNlsSJ. This data set was developed by Huang et al. (2018), and it provides the global

domestic water use, irrigation water use, livestock water use, manufacturing water use and mining water use with a spatial resolution of 0.5 degrees. We used this data set here to roughly separate the effects of human activities in the LR. For more technical information about this set of products, the readers can refer to Huang et al. (2018) and Han et al. (2019). Furthermore, detailed information on six large dams in the main stream of the LR was collected from Open Development Mekong (https://opendevelopmentmekong.net/topics/hydropower/) and Huaneng Lancang River Hydropower Inc. It mainly includes the

dates when the rivers start to be closed, when these dams were fully put into use, their dead storage capacity, their total storage capacity, and other information.





### 3. Methodologies

### 3.1 The novel proposed framework

Hydrological simulation is one of the main methodologies to quantify the contribution rate of climate change and human activities to streamflow variations; however, in the past, related studies have rarely considered the uncertainty involved in hydrological simulations (Farsi and Mahjouri, 2019). In this section, we will introduce a new quantitative framework to avoid the common phenomenon of "equifinality for different parameters" in hydrological simulation, by constructing the posterior distribution of streamflow simulations during the implementation process. The specific implementation flowchart is shown in Fig. 2. The detailed execution steps are shown as follows.

Step 1: Inspection of break points in the annual streamflow sequence; based on the result of break point test, the entire time series is divided into a natural period (before the break point) and an impacted period (after the break point).

Step 2: Sensitivity analysis of the parameters in the hydrological model.

Step 3: According to the results of the parameter sensitivity analysis, selection of the more sensitive parameters and input of the hydrometeorological data of the natural period to calibrate the hydrological model with 1000 runs.

Step 4: Selection of the parameter sets with Nash-Sutcliffe efficiency coefficients (NSE) is greater than 0.75 in 1000 simulations, input of the hydrometeorological data of the impacted period, and further calculation of the CR of CC and HAs to the streamflow change corresponding to each simulation result.

Step 5: Construction of the posterior histogram distribution (PHD) of the CR of CC and HAs (with a 5% step), and then the histogram with the highest frequency is treated as the uncertainty contribution rate interval of CC and HAs to the streamflow

change.

Step 6: The arithmetic mean of the results in the interval is treated as its true contribution rate.

In step 4, to ensure the number of streamflow simulation samples, we set the simulation results with NSE is greater than 0.75 to at least 500 times. If the setting is not met, then step 3 is repeated until the cumulative simulation times are greater than 500 times.






**Fig. 2.** Flowchart of the newly proposed quantitative framework.

### 3.2 Mann-Kendall test

In this step, the trends and break points of the hydrometeorological data are detected using the nonparametric Mann-Kendall monotonic trend test (Mann, 1945; Kendall, 1975; Gilbert, 1987) and the Mann-Kendall break point test (Sneyers, 1991),





respectively. The main consideration of using the Mann-Kendall test is that this method assumes no particular distribution for the

tested time series (Xu et al., 2018; Song et al., 2019). Significance levels of α = 0.01 and 0.05 is used in this study.

### 3.2.1 Mann-Kendall monotonic trend test

The Mann-Kendall (MK) monotonic trend test was developed by Mann (1945), Kendall (1975) and Gilbert (1987), which

has been widely used to detect the presence of an upward or downward trend of the hydrometeorological time series, and the

advantage of this test is that the time series does not need to follow a certain distribution (Hamed and Ramachandra Rao, 1998).

This method first tests whether to reject the null hypothesis ($H_0$: no monotonic trend) and accept the alternative hypothesis ($H_a$:

with monotonic trend) for a significance level of $\alpha$. The defined statistic $S$ can be calculated by the following equation:

$$S = \sum_{k=1}^{n-1} \sum_{j=k+1}^{n} sign\left(x_j - x_k\right) \tag{1}$$

where $x_k$ is the data in the order over time, $x_1$, $x_2$, …, $x_{n-1}$, which means the time series obtained at times 1, 2, …, n-1, respectively;

$x_j$ is another time series over time $x_{k+1}$, $x_{k+2}$, …, $x_n$; n is the length of the data set record; and $sign\left(x_j - x_k\right)$ is a sign function

that takes on the values of 1, 0, or -1 based on the sign of $x_j - x_k$, and its values can be calculated by the following equation:

$$sign\left(x_j - x_k\right) = \begin{cases} -1, & x_j - x_k < 0 \\ 0, & x_j - x_k = 0 \\ 1, & x_j - x_k > 0 \end{cases} \tag{2}$$

After calculating the $S$ sequence, the variance of $S$ can be computed as follows:

$$VAR(S) = \frac{1}{18}\left[n(n-1)(2n+5) - \sum_{p-1}^{g} t_p\left(t_p - 1\right)\left(2t_p + 5\right)\right] \tag{3}$$

where $n$ is the length of the time series; $g$ is the length of any given tied group and $t_p$ is the length of the data set series in the

$p_{th}$ group. Then, the defined test statistic $Z_{MK}$ can be transformed from the statistical value $S$, and the equation is as follows:

$$Z_{MK} = \begin{cases} = \dfrac{S-1}{\sqrt{VAR(S)}} & if\ S > 0 \\ = \quad 0 & if\ S = 0 \\ = \dfrac{S+1}{\sqrt{VAR(S)}} & if\ S < 0 \end{cases} \tag{4}$$

At the given significance level $\alpha$, if $-Z_{\alpha/2} \leq Z_{MK} \leq Z_{\alpha/2}$, then the $H_0$ (null hypothesis) is accepted, which means that

there is no significant trend in the time series. By contrast, a positive $Z_{MK}$ indicates that the tested time series has an upward trend,

while a negative value indicates a downward trend.





### 3.2.2 Mann-Kendall break point test

The break point of the hydrometeorological time series denotes a change from one stable state to another stable state (Xu et al., 2018). It occurs when the climate system breaks through a certain threshold. The Mann-Kendall break point test has been widely used to test break points for hydrometeorological time series, signaling when abrupt changes start (Sneyers, 1991). This test

method is used to determine the break point of the observed annual streamflow in this study. The defined statistic $UF_k$ is obtained by the following formulas:

$$UF_k = \frac{S_k - E(S_k)}{\sqrt{Var(S_k)}} \qquad k = 1, 2, ..., n \tag{5}$$

$$S_k = \sum_{i=1}^{k} r_i \qquad r_i = \begin{cases} 1 & x_i > x_j \\ 0 & else \end{cases} \qquad j = 1, 2, ..., i \tag{6}$$

where $x_i$ is the variable to be tested and $n$ is the total number of data points. The expectation $E(S_k)$ and variance

$Var(S_k)$ of the data series can be calculated as follows:

$$E(S_k) = \frac{n(n-1)}{4} \tag{7}$$

$$Var(S_k) = \frac{n(n-1)(2n+5)}{72} \tag{8}$$

$UF_k$ is a sequence of statistics calculated by arranging $x_1$, $x_2$, ..., $x_n$ in the order of time series x that obeys the standard normal distribution. Then, treating the time series x in reverse order $x_n$, $x_{n-1}$, ..., $x_1$, the above process is repeated, but by using a

reversed definition of $UB_k = -UF_k, k = n, n-1, ..., 1$. Given the significance level α (0.01 in this study), if $UB_k = -UF_k, k = 1, 2, ..., n$, no significant trend is detected, where $U_{\alpha/2}$ is the standard normal deviation. In contrast, this means that the tested sequence has a significant upward or downward trend when $|UF_k| > U_{\alpha/2}$. Then, the curves of $UF_k$ and $UB_k$ are plotted. If there is an intersection of the two curves and the trend of the data series is statistically significant, then this intersection is regarded as the break point of the data series.

After identification of the break points in the annual streamflow series, the study period is divided into a "natural period" (before the break point) and an "impacted period" (after the break point) (Bao et al., 2012; Wang et al., 2015). The "natural period" means that there is no significant increase or decrease in streamflow during this period, and it also means that relatively slow climate change is the dominant factor, and that the impact of human activities is very small during this period. Consequently, the impacted period indicates to a significant change in streamflow during this period, mostly due to factors such as the construction

of water conservancy engineering facilities, increased water consumption for irrigation, changes in land use and increased water consumption in cities and towns.





### 3.3 SWAT model

The Soil & Water Assessment Tool (SWAT) model is a semi-distributed, physical process-based hydrological model developed by the Agricultural Research Service of the United States Department of Agriculture (USDA-ARS) (Arnold et al., 1998). The

SWAT model first divides the study area into several subbasins based on DEM data, and then each subbasin is further divided into several HRUs (Hydrologic Response Units) based on land use and soil data sets. Then, streamflow generation at the subbasin scale is calculated following the principles of water balance and energy balance after inputting the meteorological data sets. Finally, the total flow of river basin exports is calculated according to the Muskingum method (Arnold et al., 2012b; Tang et al., 2019). We chose to use the SWAT model in this study because numerous published studies have proven that this model has excellent

performance in hydrological simulations across the world (Lee et al., 2018; Zhao et al., 2018a; Zhao et al., 2018b; Tang et al., 2019).

The calibration of model parameters is executed using the independent software SWAT-CUP, which was developed by Abbaspour et al. (2007). This software is freely available and provides five parameter calibration and uncertainty analysis methods. In this study, the sequential uncertainty domain parameter fitting version 2 (SUFI-2) algorithm (Abbaspour et al., 1997; Abbaspour

et al., 2004) was used to perform parameter calibration and uncertainty analysis, because this method has proven to have the advantages of shorter calculation time, ease of implementation and ability to set arbitrary objective functions (Wu and Chen, 2015; Tuo et al., 2016; Zhao et al., 2018b). The performance of the SWAT model was evaluated by the Nash-Sutcliffe efficiency coefficient (NSE) (Nash and Sutcliffe, 1970) and relative error (RE):

$$\text{NSE} = 1 - \frac{\sum_{i=1}^{N}\left(Q_{obs,i} - Q_{sim,i}\right)^2}{\sum_{i=1}^{N}\left(Q_{obs,i} - \overline{Q_{obs}}\right)^2} \tag{9}$$


$$\text{RE} = \frac{R_{sim} - R_{obs}}{R_{obs}} \times 100\% \tag{10}$$

where $Q_{obs,i}$ and $Q_{sim,i}$ are the observed and simulated streamflow, respectively; $\overline{Q_{obs}}$ is the mean value of the observed streamflow; $N$ is the total number of days or months in the calibration period; and $R_{sim}$ and $R_{obs}$ are the mean annual simulated and observed streamflow, respectively.

### 3.4 Construction of the Posterior Histogram Distribution of contribution rate

In this section, we introduce how to calculate the contribution rate (CR) of climate change (CC) and human activities (HAs) to streamflow variations and how to construct the posterior histogram distribution (PHD) of the CR to consider the uncertainty of hydrological simulations.

#### 3.4.1 Contribution rate of climate change and human activities

A schematic diagram of the attribution evaluation of streamflow changes is shown in Fig, 3. ΔQ in the figure represents the

amount of change in the observed streamflow during the impacted period based on the natural period, while $\Delta Q_{cc}$ and $\Delta Q_{ha}$ represent the amount of streamflow change caused by climate change and human activities, respectively. The total change in the annual streamflow can be calculated using the following formula:

$$\Delta \text{Q} = \Delta Q_{cc} + \Delta Q_{ha} = \overline{Q_{oi}} - \overline{Q_{on}} \tag{11}$$

where $\overline{Q_{oi}}$ and $\overline{Q_{on}}$ are the mean annual observed streamflow (m³/s) in the impacted period and natural period, respectively.

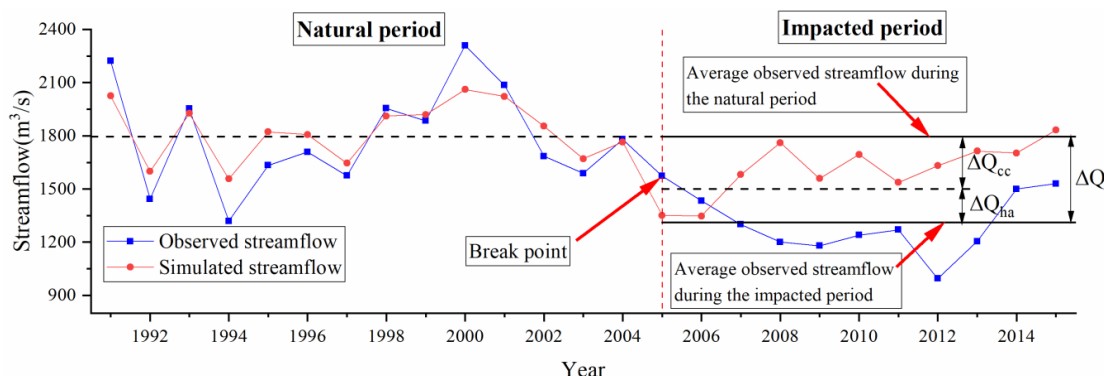

**Fig. 3.** Schematic diagram of the contribution rate of climate change (CC) and human activities (HAs) to streamflow change using SWAT modelling. (Notation: $\Delta Q$, $\Delta Q_{cc}$ and $\Delta Q_{ha}$ respectively represent the amount of streamflow change in the impacted period, the amount of streamflow change caused by climate change, and the amount of streamflow change caused by human activities.)

The hydrological and meteorological data in the natural period are input into the SWAT model, and using the SUFI-2 method to calibrate the model, a set of parameters represents the characteristics of catchment under natural conditions with less impact from human activities. Then, this set of parameters is brought back into the SWAT model using the meteorological data of the impacted period. Based on the above simulation results, the climate change induced in streamflow can be calculated as follows:

$$\Delta Q_{cc} = \overline{Q_{si}} - \overline{Q_{sn}} \tag{12}$$

where $\overline{Q_{si}}$ and $\overline{Q_{sn}}$ represent the mean simulated annual streamflow (m³/s) for the impacted period and natural period, respectively. Thus, the streamflow change induced by human activities can be calculated by the following equation:

$$\Delta Q_{ha} = \Delta Q - \Delta Q_{cc} \tag{13}$$

After the calculation of $\Delta Q_{cc}$ and $\Delta Q_{ha}$, the contribution rate of climate change and human activities to streamflow changes, which are defined as $CR_{cc}$ and $CR_{ha}$, respectively, can be estimated as:

$$CR_{cc} = \frac{|\Delta Q_{cc}|}{|\Delta Q|} \times 100\% \tag{14}$$

$$CR_{ha} = \frac{|\Delta Q_{ha}|}{|\Delta Q|} \times 100\% \tag{15}$$

Equations 12 to 15 are also applicable to quantify the contribution rate of climate change and human activities to streamflow changes on a monthly scale.

### 3.4.2 Construction of the PHD of the CR of CC and HAs

Before the construction of the PHD of the CR of CC and HAs, the sensitivity of the parameters of the SWAT model is first conducted. Based on the related published literature (Yang et al., 2008; Malagò et al., 2015; Zhao et al., 2018b) and the authors' experience, the Latin-Hypercube and global sensitivity methods were used to perform the uncertainty analysis (Abbaspour et al., 2007). The global sensitivity analysis method is the estimation of the average change in the objective function caused by the change in each parameter, and all parameters change during the whole process. A t-test was used to identify the relative sensitivity





of each parameter. Considering the influence of the snowmelt streamflow process upstream of the LR Basin on the hydrological simulation, 22 parameters were selected and the details of these selected parameters are shown in Table 2. According to the suggestion of Abbaspour et al. (2004), 500 simulations were set up to implement the sensitivity analysis. The t-stat and P-values were used to measure which parameters were more sensitive, where a larger absolute t-stat value and a smaller absolute P-value represent a higher sensitivity of a given parameter.

**Table 2** Twenty-two selected SWAT model parameters in the sensitivity analysis at Yunjinghong station

| Parameter | Description | Parameter Range |
|-----------|-------------|-----------------|
| R_CN2 | SCS runoff curve number for soil condition II | −0.2 – 0.2 |
| R_SOL_AWC | Available water capacity of each soil layer | −0.2 – 0.1 |
| A_GWQMN | Threshold depth of water in the shallow aquifer required for return flow to occur | 0 – 25 |
| V_SNOCOVMX | Minimum snow water content that corresponds to 100% snow cover | 0 – 500 |
| V_SMFMN | Minimum melt rate for snow during the year (occurs on the winter solstice) | 0 – 20 |
| V_CH_K2 | Effective hydraulic conductivity in the main channel alluvium | 0 – 500 |
| V_GW_REVAP | Groundwater "revap" coefficient | 0.02 – 0.2 |
| V_REVAPMN | Threshold depth of water in the shallow aquifer for "revap" to occur | 0 – 500 |
| V_GW_DELAY | Groundwater delay (days) | 0 – 500 |
| V_ALPHA_BF | Baseflow alpha factor (days) | 0 – 1 |
| V_SOL_BD | Moist bulk density | 0.9 – 2.5 |
| A_ESCO | Soil evaporation compensation factor | 0 – 0.2 |
| V_OV_N | Manning's "n" value for overland flow | −0.01 – 0.6 |
| R_RCHRG_DP | Deep aquifer percolation fraction | 0 – 1 |
| V_CH_N2 | Manning's "n" value for the main channel | 0.018 – 0.15 |
| R_SLSUBBSN | Average slope length | 0 – 0.2 |
| V_SMTMP | Snowmelt base temperature | −5 – 5 |
| V_TLAPS | Temperature lapse rate | −10 – 10 |
| V_SMFMX | Maximum melt rate for snow during year | 0 – 20 |
| R_SOL_K | Saturated hydraulic conductivity | −0.8 – 0.8 |
| V_SFTMP | Snowfall temperature | −5 – 5 |
| V_ALPHA_BNK | Baseflow alpha factor for bank storage | 0 – 1 |

(Notation: R_, V_, and A_ represent multiplying, replacing, and adding the corresponding parameter values, respectively, in the process of calibrating the parameters.)

Based on the sensitivity analysis results, 9 parameters with the highest sensitivity were selected to re-calibrate the model with 1000 simulations. According to the recommendations in Tuo et al. (2016) and Moriasi et al. (2007), the performance of the

hydrological simulation can be divided into four grades based on the NSE values: very good performance (0.75 ≤ NSE < 1), good performance (0.65 ≤ NSE < 0.75), satisfactory performance (0.5 ≤ NSE <0.65) and unsatisfactory performance (NSE < 0.5). According to this evaluation standard, we selected simulation results with NSE greater than 0.75 out of 1000 simulation results to construct the posterior histogram frequency distribution (PHD) of the contribution rate of climate change and human activities to





streamflow changes using the method introduced in section 3.4.1. Note that to reduce the random error caused by the number of

samples, we set the number of simulations with NSEs ≥ 0.75 to be more than 500; that is, we needed to repeatedly use Latin hypercube sampling and the SUFI-2 algorithm until the number of simulation results that met the conditions was more than 500. Then, the CR of more than 500 groups of CC and HAs to streamflow change was calculated. Finally, the posterior histogram distribution (PHD) of the CR of CC and HAs was constructed in 5% steps. At this stage, the histogram column with the highest frequency in the PHD was selected as the result of quantitative analysis, which considered the uncertainty, and the arithmetic

average of all results in the column was used as the actual value of the CR of climate change and human activities.

**3.5 Comparison of the newly developed quantification method with other two methods**

In order to evaluate the calculation accuracy of the novel framework proposed in this study to quantify the CR of CC and HAs to streamflow changes, the Budyko framework was used first. This framework was developed by Budyko (1961) and links climate variability to streamflow (Q) and actual evapotranspiration (AE) through the assumption that the long-term average annual

catchment AE is determined by the catchment average precipitation (P) and the catchment potential evapotranspiration (PET) (Liu and Liang, 2015). Over the past few decades, the Budyko framework and its variants have been widely used to conduct climate change and human activity attribution analyses of streamflow changes (Liu et al., 2017; Han et al., 2019; Xin et al., 2019). According to its theoretical assumptions, the multiyear average water balance within the catchment can be expressed as follows:

$$\Delta S = P - AE - Q \tag{16}$$

where P, Q, and AE represent the multiyear average precipitation (mm), streamflow (mm) and actual evapotranspiration (mm), respectively; $\Delta S$ (mm) is the change in the amount of water storage at the watershed scale, and it is reasonable to assume that it is equal to 0 on the multiyear average scale. According to Zhang et al. (2001), the AE can be calculated by the following formula:

$$\frac{AE}{P} = \frac{1 + \omega\left(PET/P\right)}{1 + \omega\left(PET/P\right) + \left(PET/P\right)^{-1}} \tag{17}$$

where $\omega$ is the plant-available water coefficient which is related to the vegetation type of the catchment; it is set to 0.5 in this

study.

The changes in the catchment streamflow due to climate change, which are mainly characterized by precipitation (P) and actual evapotranspiration (AE), can be expressed as follows:

$$\Delta Q_{cc} = \alpha \Delta P + \beta \Delta AE \tag{18}$$

where $\Delta Q_{cc}$ (mm) represents the streamflow changes induced by climate change; $\alpha$ and $\beta$ represent the sensitivity of streamflow

to precipitation and actual evapotranspiration, respectively; and $\Delta P$ and $\Delta AE$ are the changes in precipitation and actual evapotranspiration in the impacted period compared to the natural period, respectively. The sensitivity coefficients $\alpha$ and $\beta$ are defined as follows:

$$\alpha = \frac{1 + 2DI + 3\omega DI}{(1 + DI + \omega(DI)^2)^2} \tag{19}$$

$$\beta = -\frac{1 + 2\omega DI}{(1 + DI + \omega(DI)^2)^2} \tag{20}$$

where $DI$ is the dryness index which is equal to $PET/P$.

Through the above formulas, we can separate the contribution rate of climate change to streamflow variations, and further compare it with the calculation results of the new method proposed in this paper.





In addition to the Budyko framework, we also used the GMSWU data introduced in Section 2.2 and the reservoir dead storage capacity data to roughly separate the CR of HAs from the streamflow changes in the LR Basin. The GMSWU data set provides
five types of water withdrawals (i.e., irrigation, livestock, domestic use, mining, and manufacturing) within the period of 1970 to 2010 in the LR Basin, and it was generated by downscaling country-scale estimates of different sectoral water withdrawals from the Food and Agriculture Organization (FAO) of the United Nations AQUASTAT, which ensured its good accuracy (Huang et al., 2018). Here, AQUASTAT refers to FAO's Global Information System on Water and Agriculture (http://www.fao.org/aquastat/en/). Catchment-scale annual water use data were calculated by spatially averaging all grids within
the LR Basin, and then streamflow changes caused by each type of water use were obtained using the average annual water use value during the impacted period minus that during the natural period. As shown by Han et al. (2019) and Zhao et al. (2012), during the past two decades, dam construction has been the most significant human activity affecting the streamflow changes in the LR Basin. Therefore, in this study, we converted the dead storage capacity of 6 large reservoirs (Table 1) into units of millimeters according to their watershed control area because the impact of the reservoir on the outlet flow of the watershed can
be used as its minimum impact value on the multiyear average scale. Note that the CR of CC and HAs calculated by the above two methods was not the actual true value but rather an estimate. We used these two methods to compare with the newly proposed framework developed in this study. In addition, although the calculations of these two methods are simple, they have the following shortcomings compared with the CR of CC and HAs derived from hydrological simulation: 1) the CR of CC and HAs to streamflow variations can only be calculated on a multiyear average scale and cannot be calculated on a monthly or seasonal scale;
2) they have higher requirements for data input (e. g., data related to reservoirs); and 3) they have relatively less physical meaning compared with the streamflow simulation of the distributed hydrological model.

## 4. Results

### 4.1 Hydrological and Meteorological trends in the LR Basin

#### 4.1.1 Trends and break points of the streamflow

The results of the Mann-Kendall break point test for the annual streamflow at Yunjinghong station within the period from 1961 to 2015 are shown in Fig. 4. Since the intersection of the UF and UB curves in Fig. 4 is within the confidence intervals (of 0.05 and 0.01), the break point of the annual streamflow in the LR Basin occurred in 2005. Combined with the construction of reservoirs in the LR Basin, the construction of the Xiaowan hydropower station started in October 2004 (with total storage capacity = 15.3 km$^3$). Therefore, according to the principle of time division introduced in Section 3.1, the study period can be divided into
the natural period (from 1961 to 2004) and the impacted period (from 2005 to 2015). UF curves of the MK break point test represent the trend of the time series. As shown in Fig. 4, the observed annual streamflow at Yunjinghong station had an increasing trend before 1967, after which the annual streamflow experienced a significant decreasing trend until 2015.





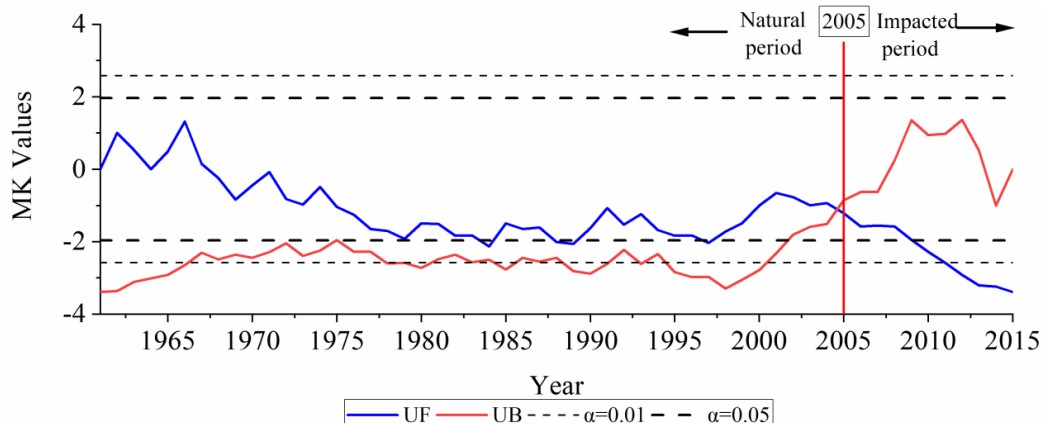

**Fig. 4.** The Mann-Kendall break point testing statistics of the annual streamflow for the Lancang River Basin from 1961 to 2015.

390        Fig. 5 shows the MK monotonic trend testing statistics of the annual and monthly streamflow for the LR Basin from 1961 to 2015. A positive Z statistic represents an upward trend in the time series, and vice versa. The annual average streamflow in the LR Basin showed a significant decreasing trend, and passed the 0.01 significance level. This decreasing trend of the annual streamflow in the LR Basin is consistent with the conclusions of Han et al. (2019). For the monthly streamflow, the streamflow from February to May showed an increasing trend, among which March and May passed the significance level of 0.05 (with the Z statistic greater than 1.96). The streamflow in the remaining months showed a decreasing trend. Except for June, the decreases in the other months all passed the 0.05 significance level. The decrease in the streamflow from August to October even passed the 0.01 significance test. Among them, the largest decrease was in August (Z statistic = -4.23). This trend of changes in streamflow during the year was mainly caused by the operation of reservoirs within the basin, because reservoirs often release flows during dry periods (from January to May) to alleviate possible droughts in the downstream areas, and they store water during wet periods (from June to October) to reduce the flood control pressure in the downstream area below the reservoir.

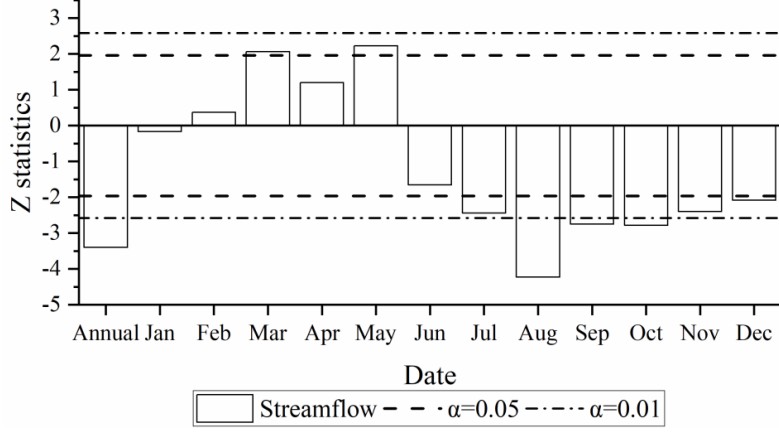

**Fig. 5.** The Mann-Kendall monotonic trend test statistics of the annual and monthly streamflow for the Lancang River Basin from 1961 to 2015.



### 4.1.2 Trends and break points of the mean areal precipitation and temperature

The time series and MK break point test results of the annual areal precipitation and mean temperature for the LR Basin from
1961 to 2015 are presented in Fig. 6. In general, changes in the annual precipitation were more complicated than changes in the
mean temperature in the LR Basin. The precipitation showed a fluctuating trend, while the mean temperature almost showed a
continuous rising trend throughout the study period.

As shown by the time series of the annual precipitation in the LR Basin in Fig. 6 (a), there was a slightly decreasing trend in
410    the LRB during the last 55 years, especially in the past 10 years, but this trend was not significant according to the result of the
MK test at the $\alpha = 0.05$ significance level. The areal annual precipitation in 1985 reached 971 mm, which was the highest in the
last 55 years. In 2009, it was 769 mm, which was the lowest value from 1961 to 2015. The MK break point test results showed
that there were 11 break points in the annual precipitation time series. Regarding the positive and negative changes in the UF
value, the annual precipitation showed a fluctuating trend from 1961 to 1966; then until 1998, the annual precipitation showed a
415    small decreasing trend (except in 1991); from 1999 to 2006, the annual precipitation experienced a small increase; and in the last
9 years (2007-2015), the annual precipitation in the study area showed a decreasing trend.

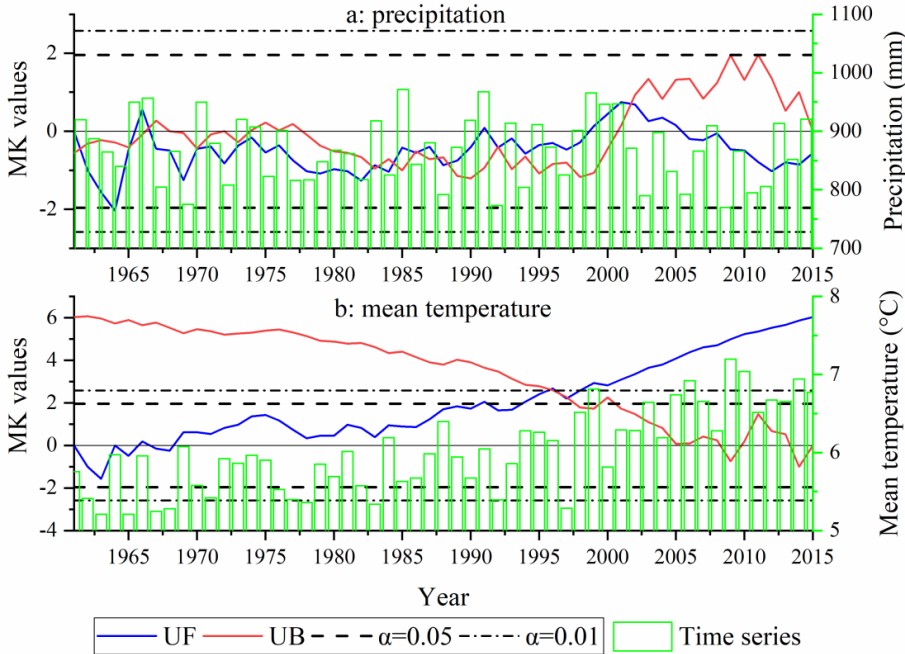

**Fig. 6.** Time series and the Mann-Kendall break point test statistics of the annual precipitation (a) and mean temperature (b) in

the Lancang River Basin from 1961 to 2015.

The time series of the annual mean temperature in the LR Basin presented in Fig. 6 (b) shows that the annual mean
temperature in the study area changed relatively smoothly before 1998. After 1998, the temperature began to rise significantly
and exceeded the significance level of 0.01. The annual mean temperature in 1963 reached 5.2 ℃, the coldest temperature in the
study period. The hottest year was 2009, during which the mean temperature was 7.2 ℃. In terms of changes in the UF value, the
mean temperature showed a fluctuating trend from 1961 to 1968, and then continued to rise until it exceeded the significance level





of 0.05 in 1991 and exceeded the significance level of 0.01 in 1998. The break point of the annual mean temperature was detected in 1997.

The MK monotonic trend test statistics of the annual and monthly precipitation and mean temperature for the LR Basin from 1961 to 2015 are presented in Fig. 7. The annual precipitation in the study area showed an insignificant decreasing trend (Z statistic = -0.55), while the annual average temperature showed a significant increasing trend (Z statistic = 6.02) and exceeded the

significance level of 0.01. The monthly change in precipitation also showed a fluctuating trend. The increasing trend of precipitation in April and the decreasing trend of precipitation in June exceeded the significance level of 0.05, while the trends of precipitation in other months were not significant (|z statistic| < 1.96). The trend of the monthly mean temperature was relatively simple. Except for the increase in the mean temperature in November, which passed the significance level of 0.05, the increasing trend of the mean temperature in all other months passed the significance level test of 0.01. This also means that the climate in

the study area has been gradually warming and drying during the past 55 years.

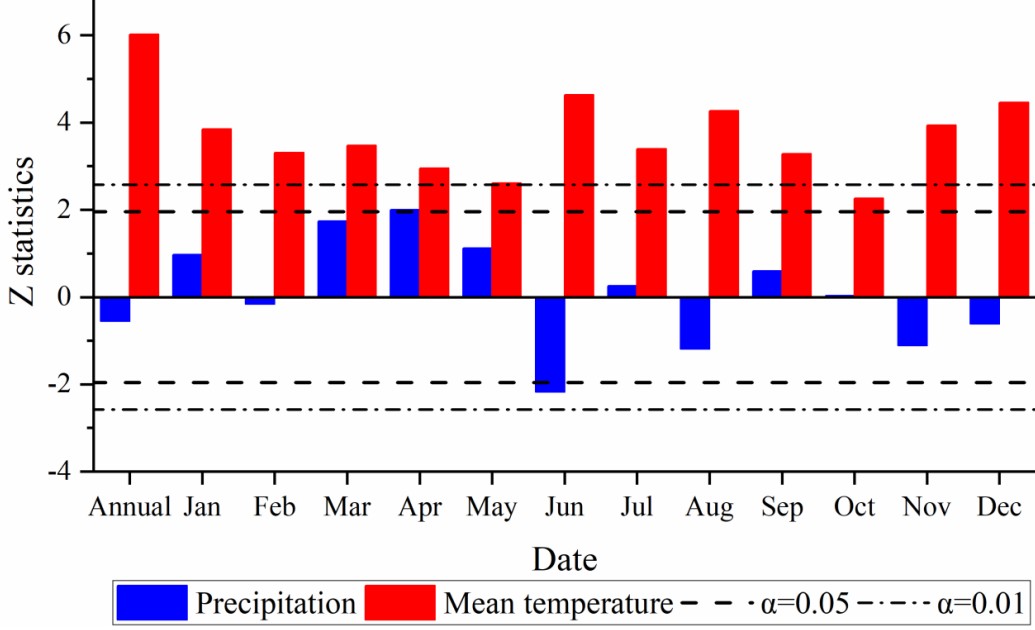

**Fig. 7.** The Mann-Kendall monotonic trend test statistics of the annual and monthly precipitation and mean temperature for the Lancang River Basin from 1961 to 2015.

### 4.2 Results of the SWAT simulations

#### 4.2.1 Sensitivity analysis of the SWAT model parameters

As described in Section 3.4.2, the sensitivity of 22 selected parameters was evaluated using the SUFI-2 and global sensitivity analysis methods. SUFI-2 performs a combined optimization and uncertainty analysis using a global search procedure and can deal with a large number of parameters through Latin hypercube sampling. The sensitivity evaluation indexes, the t-Stat and P-value, of 22 parameters are shown in Table 3. Obviously, the sensitivity ranks of the parameters calculated based on SUFI-2



showed that ALPHA_BNK has the highest sensitivity, followed by CH_K2, SOL_BD, GW_REVAP, SFTMP, CN2, SOL_K, SMTMP and ALPHA_BF, whereas the other 14 parameters have less sensitivity for the streamflow simulation. ALPHA_BNK mainly controls the baseflow process within the watershed, and this parameter has also proven to have high sensitivity in other relevant studies (Wu and Chen, 2015), especially in mountainous areas. CH_K2 and ALPHA_BF are mainly related to groundwater runoff, CN2 is the SCS runoff curve number, and these parameters all have higher sensitivity in many published

articles on the SWAT model parameter sensitivity (Wu and Chen, 2015; Zhao et al., 2018b). Other parameters with high sensitivity, such as SFTMP and SMTMP, which mainly control the snowmelt process in the basin, also indicate that snowmelt runoff plays an important role in the recharge of the LR Basin (Gao et al., 2019). Based on the above sensitivity analysis results, the top 9 parameters of the sensitivity ranking were selected for further research.

**Table 3** Basin wide sensitivity ranking calculated from 22 selected parameters using SUFI-2

| Parameter | t-Stat | P-Value | Parameter | t-Stat | P-Value |
|---|---|---|---|---|---|
| V_ALPHA_BNK | 38.1 | 0 | V_CH_N2 | 1.25 | 0.21 |
| V_CH_K2 | -10.5 | 0 | V_TLAPS | 1.19 | 0.23 |
| V_SOL_BD | 6.81 | 0 | A_ESCO | 1.12 | 0.24 |
| V_GW_REVAP | -5.0 | 0 | V_SNOCOVMX | -1.01 | 0.31 |
| V_SFTMP | -4.73 | 0 | V_PLAPS | -0.73 | 0.47 |
| R_CN2 | 4.60 | 0 | V_GW_DELAY | -0.68 | 0.50 |
| R_SOL_K | 4.34 | 0 | R_SLSUBBASN | -0.65 | 0.52 |
| V_SMTMP | 3.41 | 0 | V_OV_N | 0.44 | 0.66 |
| V_ALPHA_BF | -2.70 | 0 | V_SMFMX | 0.36 | 0.72 |
| R_SOL_AWC | -2.46 | 0.01 | A_GWQMN | 0.15 | 0.88 |
| R_RCHRG_DP | -1.74 | 0.08 | V_REVAPMN | -0.07 | 0.94 |

(Notation: V_ represents replacing the default value with the given value; R_ represents the relative change (%); and A_ represents adding the given value to the original parameter value)

### 4.2.2 Results of the SWAT simulations

As mentioned above, the 9 parameters with the highest sensitivity rankings that controlled different stages of the basin's streamflow production and flow concentration were selected to re-calibrate the model using the SUFI-2 method, and the number

of simulations was set to 2000. To reduce the influence of the initial value of the model parameters on the simulation results, during the model parameter calibration process, 1961 and 1962 were set as the warming-up period. Table 4 shows the evaluation metrics of the simulation using the SWAT model at a monthly scale with the largest NS value. For the calibration period from 1963 to 1990, the NSE and RE were found to be equal to 0.94 and -10.62%, respectively; for the validation period from 1991 to 2004, the model performance was slightly better than that in the calibration period, and the NSE and RE were 0.95 and -8.65%,

respectively. For the whole period from 1963 to 2004, the NSE (0.94) and RE (-9.97%) were also satisfactory. Fig. 8 shows the monthly observed and simulated streamflow at Yunjinghong station from 1963 to 2004 and the histogram of the mean monthly precipitation in the LR Basin. As seen from Fig. 8 (a) and Fig. 8 (b), the SWAT model can simulate the flow processes very well and almost perfectly match the observed streamflow curve. Note that the simulated streamflow overestimated the floods in individual years (1973, 1985 and 1995 in Fig.8 (b)), which might be caused by the uncertainty of the precipitation product (Han





et al., 2019). In summary, the SWAT model can better simulate the streamflow process at Yunjinghong station on a monthly scale;
        therefore, this model is considered suitable for the next part of the research.

**Table 4** Evaluation metrics, Nash-Suttcliffe Efficiency and Relative Error of the SWAT model on a monthly scale

| Period | NSE | RE (%) |
|---|---|---|
| Calibration (1963-1990) | 0.94 | -10.62 |
| Validation (1991-2004) | 0.95 | -8.65 |
| Overall (1963-2004) | 0.94 | -9.97 |

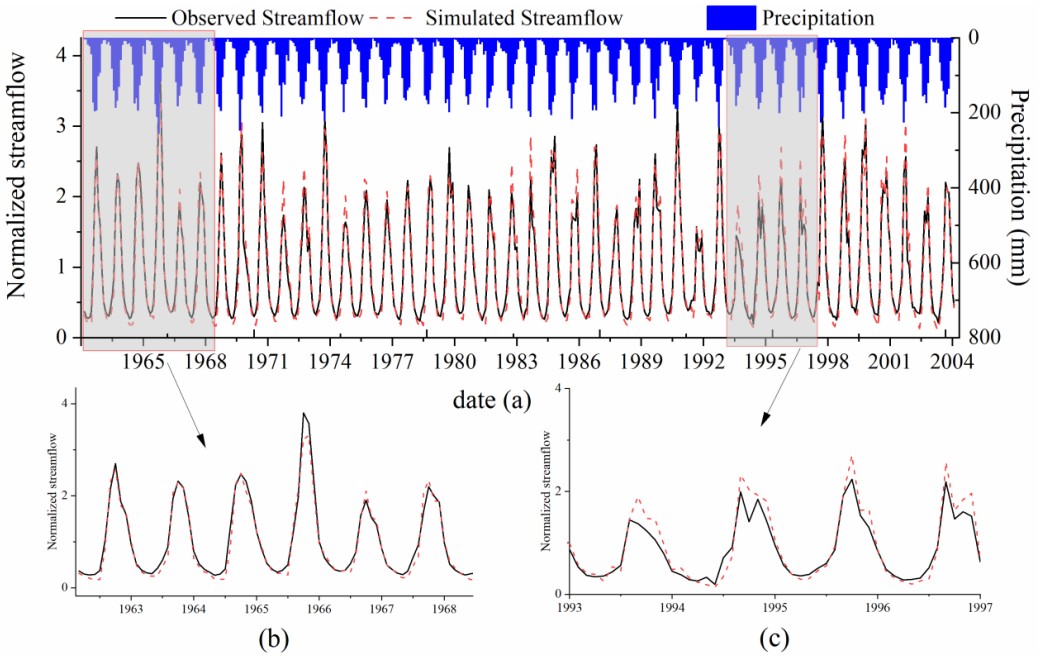

**Fig. 8.** Normalized monthly observed and simulated streamflow at Yunjinghong station for the calibration (from 1963 to 1990) and
validation (from 1991 to 2004) periods. The blue histogram shows the monthly precipitation in the LR Basin. The normalized
streamflow was calculated from the observed and simulated streamflow divided by their average values.

According to the method described in Section 3.4.2, simulations with NSEs greater than 0.75 among the 1000 simulations were
selected. Fig. 9 shows the number of simulations with $0.75 \leq NSE < 0.8$, $0.8 \leq NSE < 0.85$, $0.85 \leq NSE < 0.9$ and $0.9 \leq NSE <$
0.95 during the calibration period (1963 - 1990), the validation period (1991 - 2004) and the whole period (1963 - 2004) on a
monthly scale. In summary, there were 575 simulations with NSEs greater than 0.75 out of 1000 simulation results during the
calibration period, the validation period, and the whole period. Clearly, the NSEs of most simulation results were between 0.75
and 0.9, with 533, 537 and 533 simulations in the calibration period, the validation period, and the whole period, respectively, and
only a few simulation results had NSEs greater than 0.9. In the different periods, the model performed well in the validation period
compared with that in the calibration period, which indicated that the model has good predictive ability in the LR Basin.





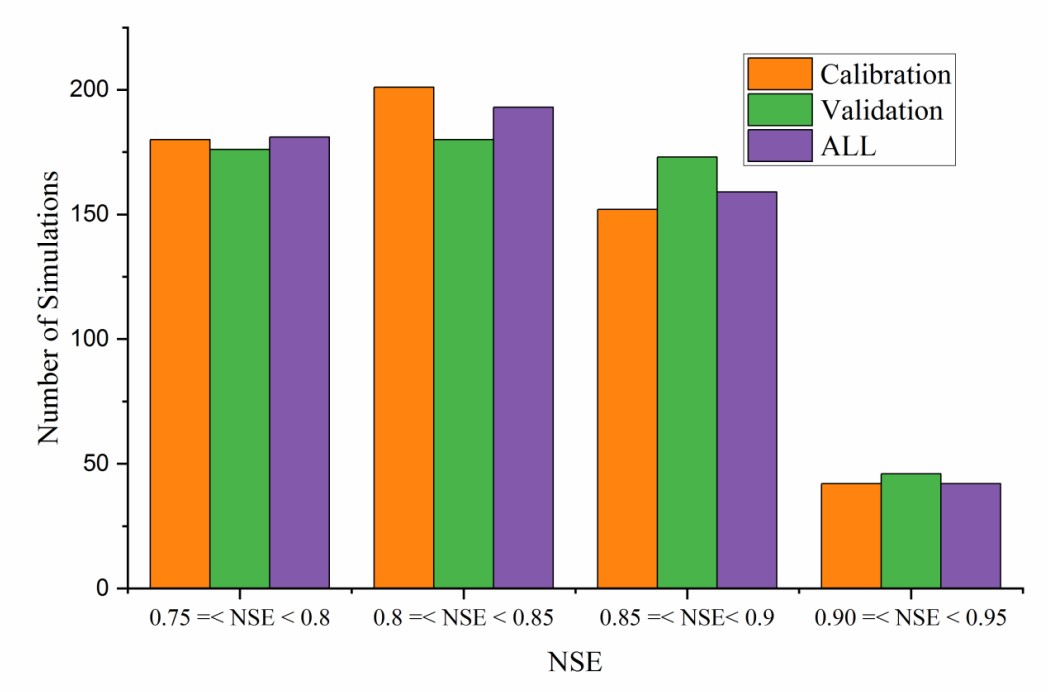

**Fig. 9**. Number of simulations with NSEs greater than 0.75 during the calibration (1963 - 1990), validation (1991 - 2004), and whole periods (1963 - 2004).

**4.3 Quantification of climate change and human activities for streamflow change considering the uncertainties**

**4.3.1 Quantification of the impacts considering the uncertainties at the annual scale**

      The 575 simulations with NSEs greater than 0.75 were selected to construct the posterior histogram frequency distribution (PHD) of the contribution rate of climate change and human activities to streamflow changes in the LR Basin. Fig. 10 shows the number of simulations of the climate change contribution rate in 5% intervals and their corresponding NS box plots. In total, 167 out of 575 simulations calculated that the contribution rate of climate change in the LR Basin to runoff reduction was 40% - 45%,

and the average NSE was 0.84. Then, 131 and 92 of the simulation results had calculated climate contribution rates of 35 - 40% and 45 - 50%, respectively. The contribution rate of climate change in other intervals had relatively few simulations. The NSE value of the climate change contribution rate between 70-75% was the largest (NSE = 0.86), but it had only 1 simulation. Therefore, when using hydrological simulations to quantify the contribution rate of climate change and human activities to the streamflow change of the watershed, not only the merits of the model performance but also the uncertainty of the model simulation should be

considered. In general, according to the results calculated by the new quantitative framework proposed in this paper, streamflow changes in the LR Basin duo to climate change accounted for 40-45% (with an average contribution rate of 42.6%), and the corresponding human activities accounted for 55-60% (with an average contribution rate of 57.4%).

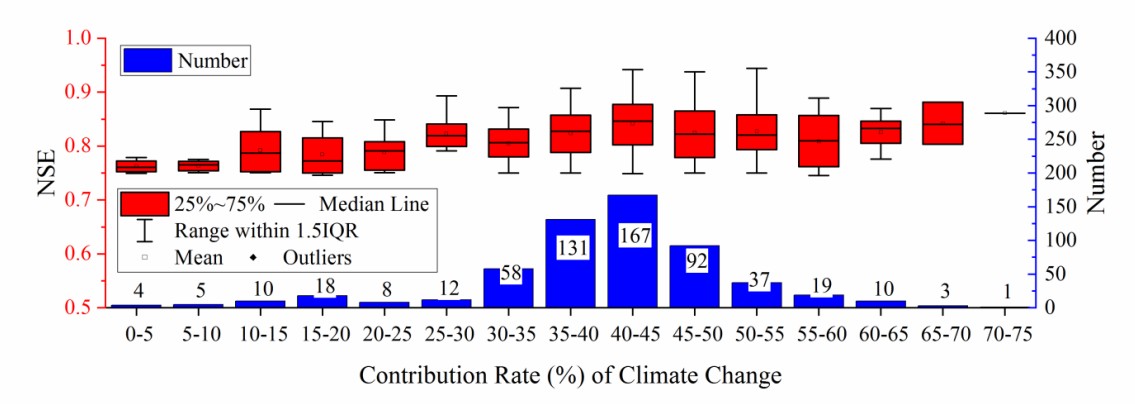

**Fig. 10**. Histogram of the number of simulations of the contribution rate (with 5% steps) of climate change to streamflow

505         reduction in the Lancang River Basin at the annual scale and corresponding Nash-Sutcliffe Efficiency box plots.

Table 5 shows the average values of the main hydrological and meteorological elements and their changes during the natural
period and the impacted period. During the impacted period, compared with the natural period, the multiyear average streamflow
decreased by 396 m³/s (86.5 mm), the precipitation decreased by 25 mm; as basin wide temperatures increased, the mean potential
evapotranspiration and temperature in the basin increased by 6.4 mm and 0.9°C. In terms of relative changes, the streamflow
decreased by 22%, but precipitation and potential evapotranspiration changed by -2.9% and 6.4%, respectively, which may
indicate that the streamflow reduction in the LR Basin was mainly caused by human activities.

**Table 5** Hydrological and meteorological elements in the natural (1963 - 2004) and impacted periods (2005 - 2015) of the LR

Basin and their changes during the two periods

| Hydrometeorological element | Streamflow (m³/s) | Streamflow (mm) | Precipitation (mm) | PET (mm) | T (°C) |
|---|---|---|---|---|---|
| Natural period | 1801.5 | 398.6 | 863.8 | 832.5 | 5.8 |
| Impacted period | 1405.5 | 312.1 | 838.8 | 885.8 | 6.7 |
| Amount of change | -396 | -86.5 | -25 | 53.3 | 0.9 |
| Relative change (%) | -22.0 | -22.0 | -2.9 | 6.4 | 15.9 |

(Notation: PET = "Potential evapotranspiration" and T = "Temperature")

**4.3.2 Quantification of the impacts considering the uncertainties on a monthly scale**

The monthly contribution rate of climate change and human activities to the changing streamflow at Yunjinghong station
was also analyzed using the new framework proposed in this study, and the results are shown in Fig. 11. In general, only June and
November had a large contribution rate of climate change, which reached 95 - 99.9% and 70 - 75%, respectively, while the
contribution rate of climate change in the other 10 months was relatively small. The trends of the streamflow and the precipitation
and mean temperature in the study area shown in Fig. 5 and Fig. 7 indicate that the streamflow in June and November showed a
decreasing trend (Fig. 5), while the precipitation in June decreased significantly (passing the significance level of 0.05), and the
temperature increased significantly (passing the significance level of 0.05) (Fig. 7). This significant decrease in precipitation and

the significant increase in temperature were the main reasons for the decrease in the streamflow in June; that is, the decrease in the streamflow in June was mainly caused by climate change. The main factors that led to the decrease in the streamflow in November were also the decrease in precipitation and the significant increase in temperature (Fig. 7). From the results of each month, the contribution rate of climate change in March and April was the smallest, reaching 10 - 15%; followed by July (15 - 20%); May, August, and September (20 - 25%); October (25 – 30%); January and February (30 – 35%); and December (45 – 50%).

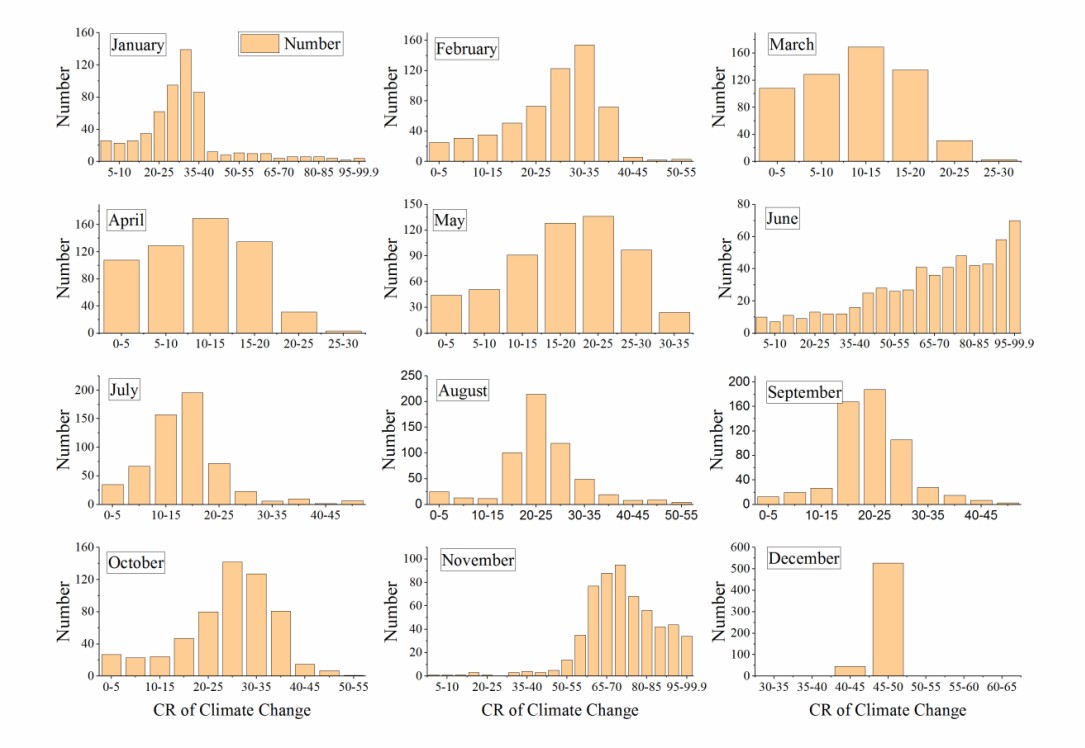

**Fig. 11**. Histogram of the number of simulations of the contribution rate (CR) (with 5% step) of climate change to streamflow reduction in the Lancang River Basin on a monthly scale.

The mean contribution rate of climate change and human activities at the monthly scale, which was calculated by averaging the contribution rates of all simulation results within the highest frequency, is displayed in Fig. 12 (left panel), and the monthly precipitation, potential evapotranspiration and runoff depth during the natural period and the impacted period are shown in the
right panel of Fig. 12. Overall, the monthly contribution rate was consistent with the annual results, and the contribution rate in a total of 10 months was mainly due to human activities that led to a decrease in the streamflow in the LR Basin. It is worth noting that the contribution rate of climate change in June reached 96%. The panel in the right of Fig. 12 shows that the precipitation in June during the impacted period was significantly reduced compared with the natural period (with a 20.2 mm decrease). At the same time, the increase in potential evapotranspiration in June was also relatively obvious (with a 9.2 mm increase). Fig. 12 (right panel) clearly shows that the streamflow in the LR Basin during the impacted period was significantly reduced compared with the





natural period in June to October, and the precipitation had little change, except in June. Therefore, we can conclude that the main reason for the decrease in the streamflow in the LR Basin was human activities, as shown in the left panel. In this study area, the main cause of the streamflow changes was mainly due to the construction of reservoirs (such as Manwan and Xiaowan), and at the same time, the water storage of these water conservancy facilities during the flood period also provides engineering support

for protecting the safety of downstream life and property. Conversely, during the dry season (from January to May), the streamflow in the impacted period showed an increasing trend compared with the natural period, and the increase in runoff during these five months was mainly due to human activities (Fig. 12, left panel), which might have been caused by the release of water from the reservoirs during the dry season. For example, in 2016, due to the influence of El Niño, the countries along the lower Mekong River all suffered severe drought. The Chinese government immediately asked the Jinghong Reservoir to release water urgently,

which effectively helped downstream countries mitigate a series of possible effects caused by drought and water shortages (Li et al., 2017b).

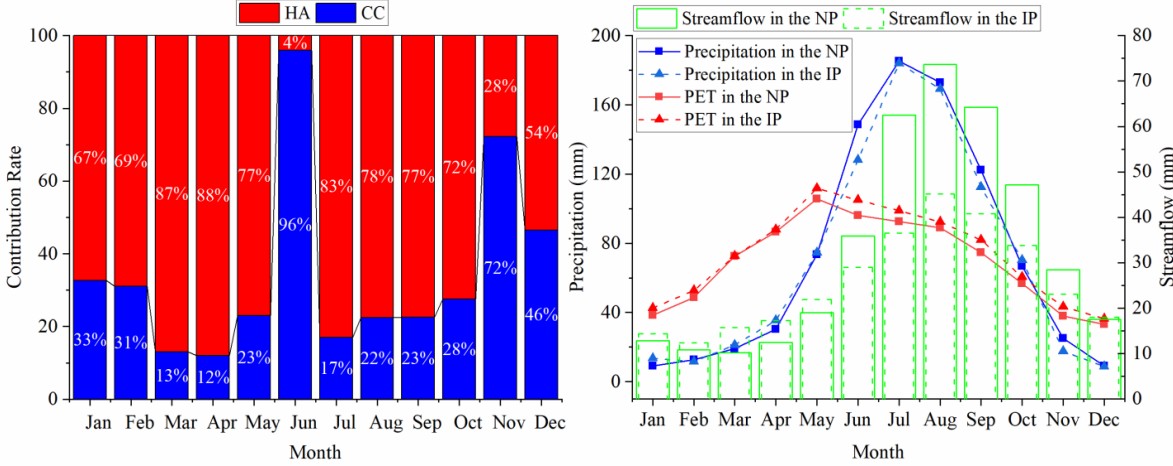

**Fig. 12**. Contribution rate of climate change and human activities to the changing monthly streamflow at Yunjinghong station (left panel), monthly precipitation, potential evapotranspiration and runoff depth during the natural period and the impacted

period in the LR Basin (right panel). (Notation: HA = "Human Activities", CC = "Climate Change", PET = "Potential evapotranspiration", NP = Natural Period and IP = Impacted Period)

### 4.4 Comparison with the other two methods

In this sub-section, the new proposed framework that considers the uncertainty of hydrological simulations was compared with the Budyko framework, five sections of water withdrawal data from the LR Basin and the equivalent streamflow depth converted

from the dead storage capacity of six large hydropower stations.

Table 6 shows the contribution rate of climate change and human activities to annual streamflow changes at Yunjinghong station, which was calculated from the Budyko framework. The actual evapotranspiration was calculated from the annual precipitation minus the annual streamflow depth. As shown in Table 6, compared with the natural period, the precipitation and streamflow depth in the impacted period showed a decreasing trend.





565                         **Table 6** Contribution rate of climate change (CC) and human activities (HA) calculated by the Budyko framework

| Time period | Precipitation (mm) | Streamflow (mm) | Actual evapotranspiration (mm) | CC (%) | HA (%) |
|---|---|---|---|---|---|
| Natural period | 863.8 | 398.6 | 463.7 | 37.2 | 62.8 |
| Impacted period | 838.8 | 312.1 | 526.6 | | |

    The precipitation decreased by 25 mm and the streamflow depth decreased by 86.5 mm. In contrast, the actual evapotranspiration showed an increasing trend, which may be related to the continuous increase in temperature in recent decades. The contribution rate of climate change and human activities to streamflow changes accounted for 37.2% and 62.8%, respectively, which was basically consistent with the results calculated by the new framework proposed in this study (the difference was 5.4%).

Fig. 13 shows the annual water withdrawals (i.e., domestic, irrigation, livestock, manufacturing and mining) in the LR Basin during the period from 1970 to 2010 and changes in the installed capacity and dead reservoir storage from 1992 to 2015. In addition to the amount of water use for irrigation, the other four types of water use withdrawals all showed an increasing trend from 1970 to 2010, with domestic water consumption increasing the most (linear slope = 0.043). The comparison between the impacted period and the natural period showed that the other four types of water consumption, except for domestic water use, all

had a larger increase in the natural period than during the impacted period. To meet the power generation needs of Southwest China and the flood control and drought resistance requirements of downstream countries, the total dead storage capacity and total installed capacity of the reservoirs from 1992 to 2015 all showed a significant increase, especially after the construction of Nuozhadu hydropower station in 2012, shown in Fig. 1.


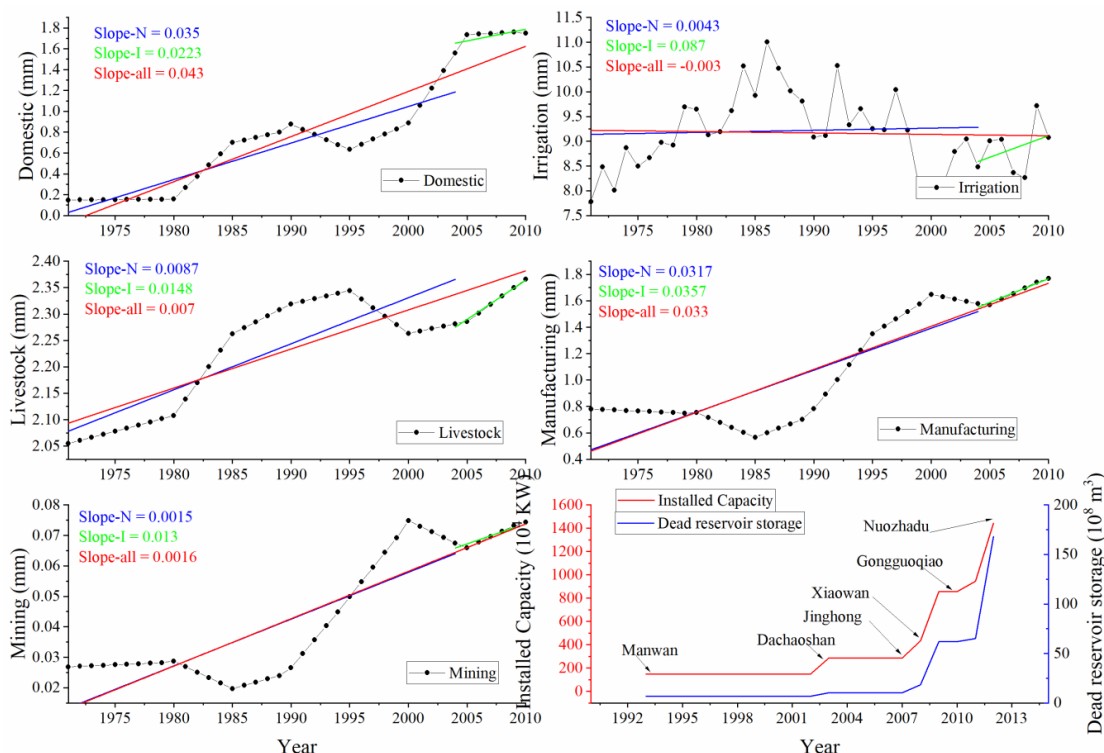

**Fig. 13**. Annual water withdrawals of the Lancang River (LR) Basin during the period from 1970 to 2010. The linear trend lines

are indicated by blue (1970-2004), green (2005-2010) and red (1970-2010), and in the last panel, the total dead storage capacity

585        and installed capacity of the LR from 1992 to 2012 are shown.

According to the method introduced in Section 3.5, the changes in the streamflow caused by human activities in the LR Basin
were separated, which mainly included the five sections of water consumption changes and the same amount of water depth as
the total dead storage capacity of the reservoir. Fig. 14 shows the contribution rate of the five types of water withdrawals by human
activities and the construction of the reservoirs to the streamflow changes in the LR Basin during the impacted period (from 2005
to 2015) compared to the natural period (from 1961 to 2004). Overall, the contribution rate of human activities to streamflow
changes was 59.91%, while that of climate change was 40.09%. This result was also consistent with the results calculated in
Section 4.3.1. Among them, the streamflow depth caused by the construction of the reservoir was reduced by -50.17 mm, which
was also the factor that had the greatest impact on streamflow compared with other human activities, and its contribution rate
reached 58.0%, while the contribution rate of the other five types of water withdrawal was relatively small. The contribution rates
of domestic, irrigation, livestock, manufacturing, and mining water withdrawals were 1.32%, -0.35%, 0.12%, 0.79% and 0.03%,
respectively, a total of 1.91%. In other words, the decrease in the streamflow in the LR Basin was mainly due to the impact of
human activities, and most of it was caused by the construction of the reservoirs.

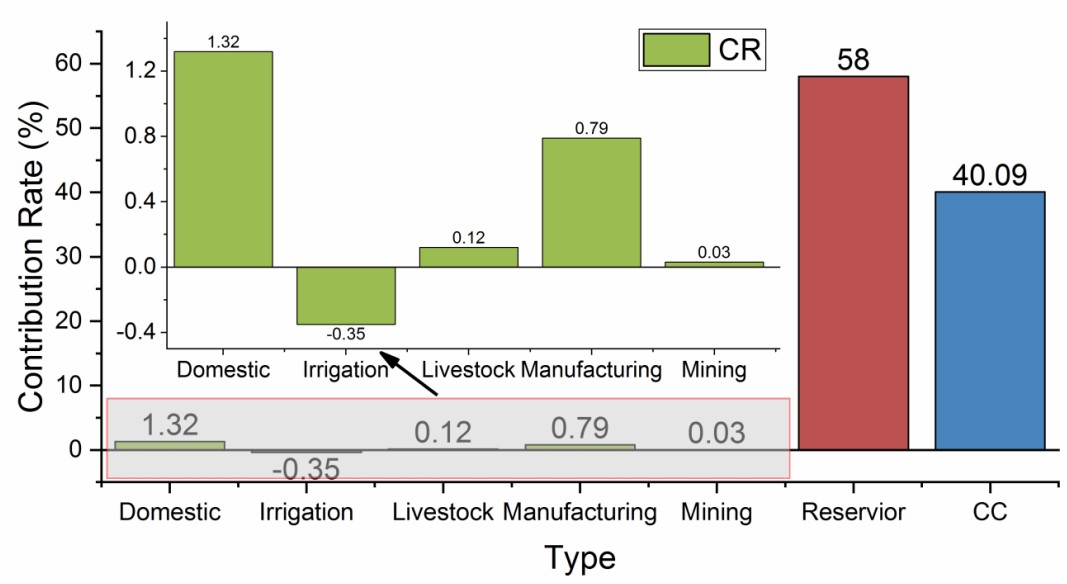

**Fig. 14**. Contribution rate (CR) of Domestic, Irrigation, Livestock, Manufacturing, and Mining water withdrawals and
reservoir construction and climate change (CC) to the streamflow changes at Yunjinghong station from 1970 to 2010.

## 5. Discussion

### 5.1 How does parameter uncertainty affect the quantitative results?

In this paper, we proposed a novel framework to quantify the contribution rate of climate change and human activities to streamflow changes considering the uncertainty of hydrological simulations. This is because the phenomenon of "equifinality for different parameters" in hydrological simulations greatly affects the quantification results. To preliminarily investigate the impact of model simulation uncertainty of the quantitative results, we selected the two simulation results with the largest NSEs in this study for analysis. The evaluation metrics and contribution rate of climate change and human activities are shown in Table 7, which shows that both simulations can simulate the monthly streamflow at Yunjinghong station in the LR Basin accurately, and the two simulations have almost the same evaluation performance. However, the attribution analysis obtained from the two hydrological simulations showed completely different results. In the first simulation result, according to the method introduced in Section 3.4.1, the streamflow changes in the LR Basin were mainly caused by climate change, but in the second hydrological simulation, the opposite conclusion was drawn, that is, human activities dominated. These were almost the same hydrological simulation results but with opposite conclusions from the attribution analysis; this was one of the reasons why we must consider the uncertainty of the model parameters in the attribution analysis of climate change and human activities using hydrological simulations. The results of Section 4.3.1 and related published studies (Han et al., 2019) in the LR Basin show that the streamflow changes in the LR Basin were mainly caused by human activities.



**Table 7** Results of the contribution rate of climate change and human activities to runoff changes with almost equal model performance (monthly) using the SWAT model

| Simulation result | Calibration | | Validation | | Overall | | Contribution Rate (%) | |
|---|---|---|---|---|---|---|---|---|
| | NSE | RE (%) | NSE | RE (%) | NSE | RE (%) | CC | HA |
| 1st simulation | 0.94 | -10.6 | 0.95 | -8.6 | 0.94 | -9.97 | 54.5 | 45.5 |
| 2nd simulation | 0.94 | -7.7 | 0.95 | -8.7 | 0.94 | -8.1 | 42.1 | 57.9 |

(Notation: CC and HA represent the climate change and human activities, respectively; NSE and RE represent the Nash-

Sutcliffe efficiency coefficient and the relative error, respectively)

Table 8 shows the values of 9 highly sensitive parameters of the two simulation results and the streamflow values simulated by the two simulations in the natural period and the impacted period. Table 8 and the calculation methods introduced in Section 3.4.1 show that the watershed streamflow reduction caused by climate change calculated by the 1st and 2nd simulation results was -

217.1 m³/s and -170.6 m³/s, respectively, which was the reason why they had opposing calculated attribution results. From the perspective of specific parameter values, the most sensitive parameter is ALPHA_BNK, which was the base flow alpha factor for bank storage (days) characterized by the bank storage recession curve. The difference between the two calibration results was not large, and this parameter mainly controlled the baseflow process, having little effect on the average annual streamflow while the difference in CH_K2 in the two calibration results was larger, at 303.87 and 106.12. This parameter represented the effective

hydraulic conductivity of the main channel alluvial layer, which meant that the larger the CH_K2 value is, the more likely the water in the main channel is lost to groundwater; accordingly, the streamflow production at the outlet of the watershed would decrease (Arnold et al., 2012b; Xu et al., 2016; Zhao et al., 2018b). This might also be one of the reasons that the first simulated streamflow (1617 m³/s) was slightly smaller than the second one (1667.9 m³/s). The SFTMP parameter, which was the temperature when precipitation was converted into snowfall, returned values for the first simulation and the second simulation as

2.69°C and -0.11°C, respectively; this meant that in the first simulation, more liquid precipitation was converted into a solid state and less streamflow was formed, which also led to a smaller simulated streamflow in the first simulation. The SMTMP parameter, which was the snow melt base temperature, was -4.13°C in the first simulation result and 3.73°C in the second simulation result. From basic physical knowledge, the SMTMP parameter in the second calibration result was more reasonable. Compared with other research results with similar terrain features in this study area, Debele et al. (2010) constructed the SWAT model in the high

altitude area of the source of the Yellow River, China, and the SMTMP value obtained was 4°C. The difference between the two simulations was not large for the set of the other parameters (SOL_BD, GW_REVAP, CN2 and SOL_K), or the parameter that controlled the baseflow (ALPHA_BF) had little effect on the average streamflow of the basin. Based on the above, the second simulation results were consistent with the calculation results of the new framework proposed in this study. Therefore, when we choose a hydrological simulation to analyze the attribution of climate change and human activities to streamflow variations, we

should clearly also consider the actual physical meaning and the uncertainties of the model parameters.






**Table 8** Values of 9 sensitivity parameters with similar simulation results and their simulated streamflow in the natural and impacted periods

| Simulation No. | 1st simulation | 2nd simulation |
|---|---|---|
| V__ALPHA_BNK | 0.84 | 0.68 |
| V__CH_K2 | 303.87 | 106.12 |
| V__SOL_BD | 1.51 | 1.93 |
| V__GW_REVAP | 0.03 | 0.003 |
| V__SFTMP | 2.69 | -0.11 |
| R__CN2 | -0.06 | -0.12 |
| R__SOL_K | 0.32 | 0.31 |
| V__SMTMP | -4.13 | 3.73 |
| V__ALPHA_BF | 0.11 | 0.77 |
| Simulated streamflow in the NP (m$^3$/s) | 1617.6 | 1667.9 |
| Simulated streamflow in the IP (m$^3$/s) | 1400.5 | 1497.3 |

(Notation: NP = "Natural period", IP = "Impacted period", R_, V_, and A_ represent multiplying, replacing, and adding the corresponding parameter values, respectively, in the process of calibrating the parameters.)

### 5.2 Land use/land cover change in the LR Basin from 1980 to 2015

In Section 4.4, the water withdrawals of domestic, irrigation, mining, livestock, and manufacturing, and in addition, dead storage capacity of constructed reservoirs as well as the impact of human activities were separated; then the impacts of human activities on streamflow changes were separated. However, human activities also influenced the land use change on rainfall-runoff characteristics. Fig. 15 shows the land use in the LR Basin in1980, 2000, 2010 and 2015. Farmland was the largest land use in the upper LR Basin, while the lower reaches were dominated by forest. Due to the high-altitude terrain in the upper reaches, unused land and glaciers were mainly distributed in this area. Table 9 shows the areas of land use types in the LR Basin in 1980, 1990, 2000, 2010 and 2015. In general, the water area of the LR Basin showed a significant reduction from 1980 to 1990, which was possibly due to the decrease in the area of glaciers due to the increase in temperature from 1980 to 1990 (Fig. 6). In contrast, the water area increased by nearly 30% from 2010 to 2015, which was mainly due to the construction of Nuozhadu hydropower station (with a total storage capacity of 22.7 km$^3$) within the basin.

**Table 9** Areas (km$^2$) of land use types in the Lancang River Basin in 1980, 1990, 2000, 2010 and 2015

| Land use type | 1980 | 1990 | 2000 | 2010 | 2015 |
|---|---|---|---|---|---|
| Farmland | 6978 | 6978 | 7034 | 6954 | 6888 |
| Forest | 33381 | 33633 | 33342 | 33417 | 33304 |
| Grassland | 42547 | 42652 | 42511 | 42500 | 42430 |
| Water | 944 | 593 | 615 | 615 | 825 |
| City | 87 | 112 | 113 | 129 | 164 |
| Unused land | 5515 | 5865 | 5837 | 5837 | 5841 |
| Permanent glacier | 724.6 | 206.7 | 247.5 | 247.5 | 243.1 |

The area of farmland in the LR Basin showed a decreasing trend during 2000-2010 and 2010-2015, which is also the main reason for the reduction in the irrigation water consumption in the basin, which is consistent with the results shown in Fig. 13. The areas of the cities all showed an increasing trend in the three periods of 1980-2000, 2000-2010 and 2010-2015 (by 29.8%,




14.1% and 27.1%, respectively), while the other three types of land use/land cover (i.e., forest, grassland, and unused land) did not change significantly in the three periods. In summary, no significant changes were found from 1980 to 2015 in the forest and grassland of the LR Basin (accounting for 37.3% and 47.5% of the total area, respectively). Although the city area has undergone significant changes, it accounts for a very small total area of the basin very (0.09%). The change in the water area was mainly due to the construction of the reservoirs, so the method used in Section 4.3 to separate the contribution of human activities to the

reduction in the streamflow in the LR Basin used is reasonable.

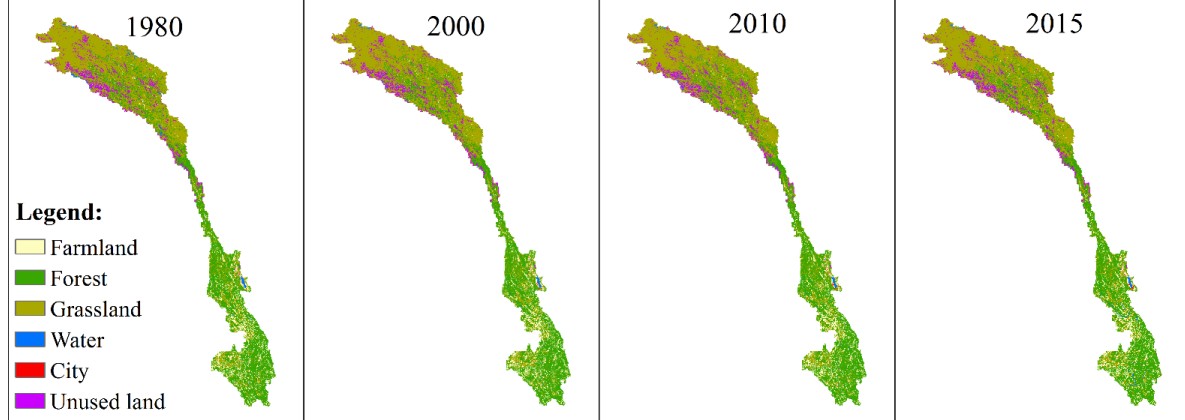

**Fig. 15**. Land use classification in the Lancang River Basin in 1980, 2000, 2010 and 2015.

### 5.3 Comparison with results of other published studies

As analyzed above, there was no particularly significant change in the precipitation and potential evapotranspiration from

1961 to 2015 in the LR Basin. Human activities mainly included the construction of reservoirs, resulting in changes in the streamflow. Attribution analysis results showed that the contribution rate of human activities was 57.6%, and the corresponding climate change was 42.4%. This result was basically consistent with Han et al. (2019), but the contribution rate of human activities was smaller than the results of Han et al. research results (95%). This may be due to the following reasons:

1) The streamflow data of different time spans were used to obtain different break points. They used streamflow data from

1980 to 2014 to obtain the break point in 2008, and this study used data from 1961 to 2015 to identify the break point in 2005.

2) Different hydrological models were used. They used the coupled routing and excess storage (CREST) model with an NSE of 0.57, while the SWAT model used in this study had an NSE of 0.94.

3) Longer series of streamflow data and simulation data were used.

As indicated by Li et al. (2017a) and (Han et al., 2019), as the streamflow data series became longer in the impacted period,

the impact of reservoir scheduling on the streamflow changes on an average scale for many years gradually decreased. Li et al. (2017a) selected Chiang Saen station, which was the nearest station to Yunjinghong station downstream of the LR Basin, for their research, and then they divided the streamflow series into three stages, the pre-impact period (1960-1991), the transition period (1992-2009) and the post-impact period (2010-2014). They concluded that the construction of the reservoirs in the LR Basin led to a decrease in the streamflow process during the flood period and an increase in the dry period, which was consistent with the

results of our study (Section 4.3.2). Their results also showed that human activities contributed 61.88% to the streamflow reduction at Chiang Saen station, which was also close to the results of our study (57.4%).


### 5.4 Applicability and uncertainty of the proposed framework

A new quantitative framework for calculating the contribution rate of climate change and human activities to watershed streamflow variations was proposed in this study, and it was successfully applied to the Lancang River Basin with relatively accurate results. From our perspective, this method can effectively avoid the phenomenon of "equifinality for different parameters" that may exist in the use of hydrological simulation methods to quantify the CR of CC and HAs. At the same time, we also believe that this framework can be applied to other watersheds based on the following aspects. First, in this study, we used the Budyko framework and the sectional water withdrawal data within the basin to compare with the new framework. Second, the results of the comparison with published research on the Lancang River also proved that the framework has good accuracy and applicability. Third, in the process of comparing with the new framework, we fully considered the impact of various human activities within the study area, including five types of water withdrawals (i.e., irrigation, livestock, living, mining, and manufacturing), the impact of reservoir storage and the land use/land cover change. Of course, due to the highly nonlinear relationship between the parameters of the hydrological model, we suggest that readers ensure that the selected simulation results with NSEs greater than 0.75 are large enough when applying the novel framework in other research areas (this study had 500 simulations). Although this quantitative framework has been proved to have good applicability in the Lancang River Basin, it may also have certain limitations. First, since there is only one break point in the mean annual streamflow sequence of the Yunjinghong station in the LR basin, there may be multiple break points in other study areas. If there exists more than one break points, the results of the break point inspection of the mean annual precipitation and potential evapotranspiration in the study area, as well as the construction time of large reservoirs in the study area, should be considered (Dey and Mishra, 2017). Finally, a unique break point is selected to divide the research time series into a natural period and an impacted period, and then the quantitative framework proposed in this study can be applied. Second, because the SWAT model has good applicability at the Yunjinghong station in the LR Basin, it can meet the 500 best simulation requirements set by the framework proposed in this study, but the hydrological model may have different applicability in different research areas Therefore, the application of this framework in other research areas may have limitations, which need to be further verified. Third, because this study uses the parameter combinations obtained by the natural period to input the meteorological element data of the impacted period for calculation, this may also bring uncertainty to the calculation results, which is usually called "transferability" (Fu et al., 2018).

Although the new quantitative framework proposed in this study considers the uncertainties in hydrological simulations, the framework is still based on traditional hydrological simulation methods to separate the contribution rate of climate change to streamflow change, and then to deduce the contribution rate of human activities. Therefore, inevitably, there are still uncertainties in the calculation process. For example, the construction of large-scale reservoirs and changes in land use/land cover (urbanization, etc.) are important factors that alter the climatic state of a local region, specifically in that they change the temporal and spatial distribution characteristics of local regional hydrometeorological elements (Degu et al., 2011; Li et al., 2017c). This change in meteorological elements was regarded as part of the impact of climate change in this study; however, it was also caused by human activities (reservoir construction). On the other hand, there are uncertainties in the division of the natural period and the impacted period in this study, which assumed that the impact of human activities on streamflow changes in the natural period was negligible; however, there were almost no periods within a watershed that were completely unaffected by human activities, and the impact of human activities on streamflow variations in the natural period was ignored in these studies. In addition, our study selected the NSE and RE as the objective function to calibrate the SWAT model, which may also bring uncertainties in the quantitative results. As indicated by Gupta et al. (2009) and Gupta and Kling (2011), using the NSE as an objective function to calibrate a hydrological





model may tend to underestimate the peak streamflow. Although the contribution rate in our study was calculated by the average streamflow over multiple years, it still brought a given amount of uncertainty to the quantitative results. Therefore, follow-up research should strengthen the optimization of the objective function and benefit from field investigation of the actual meaning of the parameters. Since the impacts of climate change and human activities on the hydrological processes of the watershed are complicated and interconnected, it is still a challenge to completely separate the impacts of climate change and human activities

on streamflow variations (Xin et al., 2019). Further consideration should be given to quantify the impact of specific human activities, such as land use change and water withdrawal, and then to separate the impact of climate change and human activities on streamflow changes as completely as possible.

## 6. Conclusions

In this study, we proposed a new framework that considered the uncertainties of model simulations to quantify the

contribution rate of climate change and human activities to streamflow changes. This framework was developed based on the posterior histogram frequency distribution (PHD) of the contribution rate of climate change and human activities. Then, we selected the Lancang River (LR) Basin for the case study. Over the past three decades, after the construction of the Manwan Reservoir in 1987, six large reservoirs were constructed within the basin before 2014. The streamflow process in the watershed also has significant changes on multiyear average and monthly scales. The Mann-Kendall monotonic trend test and the Mann-

Kendall break point test were used to test the trend and identify the break point of the annual streamflow data at Yunjinghong station within the period of 1961 to 2015. Then, the available period was divided into the natural period (before the break point) and the impacted period (after the break point). Afterwards, the SWAT model and the SUFI-2 method were used to construct the Posterior Histogram Distribution (PHD) of the contribution rate of climate change and human activities. Finally, the Budyko framework and the basin wide gridded monthly sectoral water use (GMSWU) data set were used to compare with the newly

proposed framework. The main conclusions of this study are as follows:

1) The new proposed framework can be used to quantify the contribution rate of climate change and human activities in the LR Basin which can fully solve the local optimal solution for hydrological simulation parameters in current related studies. The results of comparison using the Budyko framework and Gridded Monthly Sectoral Water Use (GMSWU) data set also showed that the new framework has high accuracy (the error range is within 6%).

2) The break point of the streamflow sequence during 1961-2015 at Yunjinghong station was identified in 2005. The streamflow significantly decreased (~ -22%) after 2005 compared with that of the natural period (1961 - 2004), which was mainly due to the construction of the Xiaowan Reservoir in October 2004. Significantly reduced streamflow in the flood period and significantly increased streamflow during the dry period also occurred, which was mainly due to the capacity adjustment of the constructed reservoirs. The trend test results also showed that from 1961 to 2015, the annual streamflow in the LR Basin showed a significant

decreasing trend at the $\alpha = 0.01$ significance level, precipitation showed a nonsignificant decreasing trend, and mean temperature showed a significant increasing trend at the $\alpha = 0.01$ significance level.

3) The quantification results calculated using the new proposed framework showed that, on an annual scale, compared with the natural period of 1961 – 2004, the Contribution Rate of Climate Change and Human Activities (CR of CC and HAs) were 40 - 45% (with an average contribution rate of 42.6%) and 55 – 60% (with an average contribution rate of 57.4%), respectively. The

CR of climate change and human activities derived from the Budyko framework were 37.2% and 62.8%, respectively, and the





error between the two calculation results was 5.4%. The CR of human activities calculated using the GMSWU data and the reservoirs dead capacities was 58.0%, which also proved that the new proposed framework in this study can be used in the LR Basin.

4) Quantitative analysis results on a monthly scale in the LR Basin showed that, except for June and November, streamflow
changes in other months were caused by human activities. Further analysis showed that the streamflow in June during the impacted period decreased by 6.9 mm compared with that in the natural period, while the precipitation and potential evapotranspiration decreased and increased by 20.2 mm and 8.83 mm, respectively; the streamflow decreased by 5.34 mm in November, while the corresponding precipitation and potential evapotranspiration changed by -7.43 mm and 5.52 mm, respectively.

In summary, this study provides a new calculation framework that considers the uncertainty of hydrological simulations to
quantify the contribution rate of climate change and human activities to streamflow changes. The results of this case study also provide a reference for understanding the dominant factors of streamflow changes in the Lancang River Basin and improving water resource management measures for the transboundary Lancang-Mekong River Basin. Of course, this new proposed framework also needs to be applied and verified in more research areas. In addition, this framework only considers the dual impacts of climate change and human activities. However, in practical applications, water resource decision makers are more
willing to understand the specific impacts of human activities such as irrigation water and land use changes. Therefore, in future research, efforts should be made to expand the framework to quantify the contribution rates of individual items of climate change and human activities.

*Acronyms:*

AE – Actual Evaporation; CC – Climate Change; CGDPA – China Gauge-based Daily Precipitation Analysis; CMA – China
Meteorological Administration; CR – Contribution Rate; CREST - Coupled Routing and Excess Storage; DEM - Digital Elevation Model; GMSWU - Global gridded Monthly Sectoral Water Use data set; HA – Human Activity; HRU – Hydrologic Response Unit; HWSD V1.2 – Harmonized World Soil Database Version 1.2; LR – Lancang River; NSE – Nash-Sutcliffe Efficiency coefficient; PET - Potential Evapotranspiration; PHD - Posterior Histogram Distribution; RE – Relative Error; SRTM - Shuttle Radar Topography Mission; SUFI -2: Sequential Uncertainty Fitting Procedure version 2; SWAT – Soil & Water Assessment
Tool; USDA - US Department of Agriculture Research Service.

**Author contributions**

Xiongpeng Tang Proposed the research framework and wrote the draft manuscript. Guobin Fu and Chao Gao Improved the language of the manuscript. Guoqing Wang and Jianyun Zhang provided the data for this study. Cuishan Liu, Yanli Liu, Zhenxin Bao, Silong Zhang and Junliang Jin Jointly processed the data, helped build the hydrological model.

**Competing interests**

The authors declare that they have no conflict of interest.



**Code and data availability**

The observed streamflow data from the Information Center of the Ministry of Water Resources and the local water resources management department are available upon request from the corresponding author (gqwang@nhri.cn) and the first author

(xptanghhu@163.com). The observed precipitation, temperature, wind speed, relative humidity data sets can be downloaded from https://data.cma.cn/.

**Acknowledgements**

This research was financially sponsored by two research programs in China: (1) the National Key Research and Development Program (No. 2018YFC1508104), (2) the National Natural Science Foundation of China (No. 92047301,52079079,51879163),

and, (3) the Second Tibetan Plateau Scientific Expedition and Research Program (No. 2019QZKK0203).

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
