# Peer review of "Attribution of climate change and human activities to streamflow variations with a posterior distribution of hydrological simulations"

_Hydrology and Earth System Sciences, 2021_

## Author Comment (AC1)

**Revision notes for "Attribution of climate change and human activities to streamflow variations with a posterior distribution of hydrological simulations"**

**(hess-2021-528)**

We would like to thank the editors and reviewers for the constructive feedback. We appreciate the valuable and thoughtful comments, which have certainly helped to improve the presentation and quality of our manuscript. We have updated our paper according to your comments and the detailed responses to the comments are described as follows.

Answers to the reviewers in blue.
Modifications of the manuscript in orange.

**To Reviewer #1:**

**General Comments:**

Runoff change attribution is an interesting research topic. In this manuscript, the authors proposed a novel framework to quantitatively evaluate the contributions of climate change and human activities to runoff changes in the Lancang River Basin. It provides an optional method for attribution of runoff changes on watershed scale, which is valuable. Generally, the manuscript is well organized and presented interesting results, but the description of the innovation of the study should be enhanced.

**Responses to comments one by one:**

1. The authors did not present the advantages of the proposed method clearly. Please clarify what's the innovation of this study.

**Response:** Thanks very much for your comments. Hydrological simulation is the main method to quantify the contribution rate (CR) of climate change (CC) and human activities (HAs) to the change of streamflow in the basin, and there is a phenomenon of "equifinality for different parameters" in hydrological simulation. Therefore, the main scientific contribution (innovation) of our study is to reduce the impact of "Equifinality" in using hydrological simulation to quantify the CRs of the CC and HAs. The accurate quantitative results have important scientific value for the development of effective water resources utilization and ecological flow regulation policies in the basin. In the abstract, we modified a part of the content to strengthen the innovation of this study in Page 1 Line 14-16, which as follows:

In our new quantitative framework, the uncertainty of hydrological simulations is first considered to quantify the impact of "equifinality for different parameters", which is common in hydrological simulations.

2. Line 171. I suggest the authors clarify the reasons for selecting the dataset in the study (GMSWU). Is it because of the higher accuracy of the data set, difficult to obtain other field data or some other reasons?

**Response:** Thank you very much for your comments. You are right that the main reasons for selecting the dataset in the study (GMSWU) are that it is difficult to collect water withdrawal data related to human activities in the Lancang River basin. Therefore, after referring to some published related literatures (Han et al., 2019, Huang et al., 2018), we choose the GMSWU dataset for related calculations. We also added a description of the reasons for choosing the GMSWU dataset in the revised manuscript. The added content (Page 6, Line 180-181) is as follows:

This dataset is used in this study because it is difficult to collect water withdrawal data related to human activities in the LR Basin, and this dataset has been successfully applied in this basin in other studies (Han et al., 2019).

**References:**

Han, Z., Long, D., Fang, Y., Hou, A., Hong, Y., 2019. Impacts of climate change and human activities on the flow regime of the dammed Lancang River in Southwest China. J Hydrol, 570: 96‑105.

Huang, Z., Hejazi, M., Li, X., Tang, Q., Vernon, C., Leng, G., Liu, Y., Döll, P., Eisner, S., Gerten, D., Hanasaki, N., Wada, Y., 2018. Reconstruction of global gridded monthly sectoral water withdrawals for 1971–2010 and analysis of their spatiotemporal patterns. Hydrol. Earth Syst. Sci., 22(4): 2117‑2133.

3. Line 181: It is not "avoid the common phenomenon of 'equifinality for different parameters' in hydrological simulation", in fact, the phenomenon of "equifinality for different parameters" in hydrological simulation is unavoidable. Suggest to revise it to "reduce the impact of "equifinality for different parameters" in hydrological simulation".

**Response:** Thanks very much for your suggestion. We acknowledge that " equifinality for different parameters" in hydrological simulation cannot be completely avoided. We run the model 1000 times, which is not to reduce the equifinality, but to quantify them and better to separate the CC and HA impacts on streamflow. The phenomenon of "Equifinality" in hydrological simulation means that different parameter combinations have similar objective function values in the simulation process, which was originally introduced to hydrology by Beven (1993). Our study is to solve the possible impact of "Equifinality" in hydrological simulation on the quantitative results, and proposes a new quantitative framework to reduce this impact. Therefore, we also compared the effect of "Equifinality" on the quantification results in Section 5.1 (5.1 How does parameter uncertainty affect the quantitative results?), and found that the quantization

results of the two simulations with the same objective function value are completely opposite. In the revised manuscript, we have corrected this descriptive error you pointed out, and the modified content is as follows (Page 8, Line 191-193):

In this section, we will introduce a new quantitative framework to reduce the influence of the common phenomenon of "equifinality for different parameters" in hydrological simulation on the quantitative results, by constructing the posterior distribution of streamflow simulations during the implementation process.

References:
Beven, K., 1993. Prophecy, reality and uncertainty in distributed hydrological modelling. Adv Water Resour, 16(1): 41-51.

4. Line 245, If multiple break points are detected for the annual runoff time series, how to select the break points and divide the whole period to natural period and impacted period? The authors should clarify this in the text.

**Response:** Thank you very much for your useful comments. In fact, for the time period division of the runoff series, we selected three break point testing methods for cross-validation (Fig. 1). As shown in Fig. 1, all three methods indicated that the annual runoff sequence of the Yunjinghong station changed abruptly in 2005. Therefore, for brevity, we only show the result of the MK break point test in the paper.
As described in section 3.2 of Dey et. al (2017), if there are multiple break points in the annual streamflow time series, multiple points need to be screened according to the types of main human activities within the study area. In this study, there is only one break point in the annual streamflow time series. In addition, Dey et. al pointed out that "*Historical records like time of construction of dam/diversion structures also help in selection of representative change point.*" In the Lancang River Basin, the construction of the Xiaowan Hydropower Station began in December 2004, so it is reasonable to choose 2005 as the break point in this study. Likewise, in Section 5.4 of our revised manuscript, we added some content to explain how to select a single break point when multiple break points exist (Page 34, Line 752-756).

First, if there are multiple break points in the annual streamflow sequence, then when selecting the unique break point, it is necessary to consider the abrupt change points of the time series of other meteorological elements (precipitation, temperature, etc.) in the basin. At the same time, the impact of strong human activities (reservoir construction, large-scale water transfer project construction, etc.) on the abrupt change of streamflow in the basin should also be considered (Dey and Mishra, 2017).

[Figure]

**Fig. 1.** Results of MK test, Moving t test and LePage test of the annual streamflow at Yunjinghong station from 1961 to 2015

**References:**

Dey, P., Mishra, A., 2017. Separating the impacts of climate change and human activities on streamflow: A review of methodologies and critical assumptions. J Hydrol, 548: 278-290.

5. Line 290-303, The proposed method of quantifying the contributions of climate change and human activities to watershed runoff changes may not guarantee that the sum of the contributions equals to 100% (e.g., equations 6 and 7). This is to say, there is an intersection between climate change and human activities. It is recommended that the authors should give explanations about this issue in the discussion section.

**Response:** Thanks very much for your comments. We admit that there are uncertainties in the method to separate the contribution rate of climate change and human activities to streamflow change, because it does not consider the mutual influence and overlap between the two impact factors. However, since the impacts of climate change and human activities on the hydrological process of the basin are very complied and interacted, it is still a challenge to completely separate the impacts of climate change and human activities on streamflow variations. Therefore, in this revised manuscript, we have revised the discussion of uncertainty about this method in Section 5.4.

   The revised discussion in Section 5.4 is described in Page 33 - 34, Line 738 - 764, which is as follows:

   A new quantitative framework for calculating the CR of CC and HAs to watershed streamflow variations was proposed in this study, and it was successfully applied to the

LR Basin with relatively accurate results. From our perspective, this method can effectively reduce the influence of the "equifinality for different parameters" that may exist in the use of hydrological simulation methods to quantify the CR of CC and HAs. At the same time, we also believe that this framework can be applied to other watersheds based on the following aspects. First, in the section 4.4, the Budyko framework and sectional water withdrawal data within the basin were used to compare with the new framework. Second, the results of the comparison with published research on the LR Basin (Han et al., 2019) also proved that the framework has good accuracy and applicability. Third, in the process of comparing with the new framework, we fully considered the impact of various HAs within the study area, including five types of water withdrawals (i.e., irrigation, livestock, living, mining, and manufacturing), the impact of reservoir storage and the land use/land cover change. Of course, due to the highly nonlinear relationship between the parameters of the hydrological model, we suggest that readers ensure that the selected simulation results with NSEs greater than 0.75 are large enough when applying the novel framework in other research areas (this study had 500 simulations). It is undeniable that this method still has certain uncertainties and limitations when it is applied to other watersheds. First, if there are multiple break points in the annual streamflow sequence, then when selecting the unique break point, it is necessary to consider the abrupt change points of the time series of other meteorological elements (precipitation, temperature, etc.) in the basin. At the same time, the impact of strong human activities (reservoir construction, large-scale water transfer project construction, etc.) on the abrupt change of streamflow in the basin should also be considered (Dey and Mishra, 2017). Finally, a unique break point is selected to divide the research time series into a natural period and an impacted period, and then the quantitative framework proposed in this study can be applied. Second, because the SWAT model has good applicability at the Yunjinghong station in the LR Basin, it can meet the 500 best simulation requirements set by the framework proposed in this study, but the hydrological model may have different applicability in different research areas. Therefore, the application of this framework in other research areas may have limitations, which need to be further verified. Third, because this study uses the parameter combinations obtained by the natural period to input the meteorological element data of the impacted period for calculation, this may also bring uncertainty to the calculation results, which is usually called "transferability" (Fu et al., 2018).

**References:**

Dey, P., Mishra, A., 2017. Separating the impacts of climate change and human activities on streamflow: A review of methodologies and critical assumptions. J Hydrol, 548: 278–290.

Fu, G., Charles, S.P., Chiew, F.H., Ekström, M., Potter, N.J., 2018. Uncertainties of statistical downscaling from predictor selection: Equifinality and transferability. Atmospheric research, 203: 130–140. DOI: 10.1016/j.atmosres.2017.12.008

Han, Z., Long, D., Fang, Y., Hou, A., Hong, Y., 2019. Impacts of climate change and human activities on the flow regime of the dammed Lancang River in Southwest China. J Hydrol, 570: 96–105.

6. Line 475, Why the authors present the normalized runoff process of the Yunjinghong station? Please clarify the reasons.

**Response:** Thanks a lot for your comments. Because the Lancang-Mekong River is a cross-border river in Southeast Asia, the runoff data was standardized in accordance with the requirements of the Information Center of the Ministry of Water Resources of the People's Republic of China, the provider of the runoff data in this study. In the revised manuscript, we have also made relevant explanations. The added explanatory content is on Line 482-484, Page 20, as shown below.

   According to the requirements of the Information Center of the Ministry of Water Resources, the data provider, this study standardized the observed and simulated runoff curves of the Yunjinghong station.

7. Section 4.4, In this section, the authors compared the results of the new quantitative framework proposed by the manuscript with two simpler methods. What's the advantage of the proposed method over the two methods? please clarify.

**Response:** Thanks very much for your comments. In this study, we used two methods to compare with the calculation results of the novel quantitative framework proposed in this study. We have discussed the shortcomings of these two simple methods compared with the quantitative analysis framework of this study in Section 3.5 to highlight the innovation of this paper. The revised content is described in Page 16, Line 382 – 386, which is as follows:

   It should be pointed out that here we use two seemingly simpler methods to verify the computational results of the new framework proposed in this study. However, this does not reduce the innovation of this study, as the new framework has the following significant advantages over the other two methods: 1) The new framework can perform quantitative calculations on the annual and monthly scales; 2) It has relatively less data requirements; 3) It has a more explicit physical meaning.

8. Fig 15, (1) How did you produce land use maps? Please clarify the satellite data that you used. (2) The presented land use maps in 1980, 2000, 2010 and 2015 look like similar, no obvious land use changes can be identified. (3) Figure 15 shows 6 types of land use while table 9 shows 7 types. They should be consistent.

**Response:** Thank you very much for your comments. (1) The land use dataset used in this study is downloaded from the Geographic Information Monitoring Cloud Platform (http://www.dsac.cn/), with a spatial resolution of 30 meters, providing land use information in the whole range of China. This study uses ArcGIS software to intercept the land use data in the Lancang River Basin. (2) In the revised manuscript, we have replaced the Figure 15, and only put the land use data of the Lancang River Basin in

2015 for display. (3) In Figure 15, the land use information we show is the first-level type information of the Lancang River Basin, while the permanent glacier in Table 5 is second-level type and belong to water in Figure 15.

For the above three comments, we have also made corresponding revisions in the revised manuscript. The modified content is as follows:

(1) Page 7, Line 173-176

In this study, to analyze the land use change in the LR during the historical period, we collected five periods of land use data in the 1980s, 1990s, 2000s, and from 2010 to 2015, and this data set was downloaded from the Geographic Information Monitoring Cloud Platform (http://www.dsac.cn/), with a spatial resolution of 30 meters.

(2) Page 32

[Figure]

**Fig. 15**. Land use classification in the Lancang River Basin in 2015.

(3) We added an explanation for this comment in Page 31, Line 704.

(Nation: Permanent glacier in Table 5 is second-level type which belong to Water)

**Technical corrections:**

9. In addition, the format of the listed references is not uniform. For example, some references have DOI information, but some do not (Lines 831, 846, 915, etc).

**Response:** Thanks very much for your suggestion. We have performed a detailed review of the reference format in the revised manuscript, and added missing DOIs to ensure a consistent format for all the listed references.

10. If some nouns have been abbreviated in the manuscript, please use the abbreviation after the first occurrence and keep consistent throughout the whole text. For example, in line 13, please replace "contribution rate" with "CR". The same problem exists in other noun expressions in the manuscript. Please review the full text in detail and make consistent revisions (Line 16: human activities, Line 19: Lancang River Basin, etc.).

**Response:** Thank you very much for your comments. We have carefully checked all the abbreviations of nouns throughout the manuscript to ensure that they are full names the first time they occur, and all abbreviations thereafter.

11. Table 5, The titles of the hydrometeorological elements in the table should be consistent. At the moment, some of them are full names, and some of them are abbreviations.

**Response:** Thanks very much for your suggestion. We have modified Table 5 in the revised manuscript.

The revised Table 5 is as follows (Page 23):

Table 5 Hydrological and meteorological elements in the natural (1963 - 2004) and impacted periods (2005 - 2015) of the LR Basin and their changes during the two periods

| Hydro-meteorological element | Streamflow ($m^3/s$) | Streamflow (mm) | Precipitation (mm) | Potential evapotranspiration (mm) | Temperature (°C) |
|---|---|---|---|---|---|
| Natural period | 1801.5 | 398.6 | 863.8 | 832.5 | 5.8 |
| Impacted period | 1405.5 | 312.1 | 838.8 | 885.8 | 6.7 |
| Amount of change | -396 | -86.5 | -25 | 53.3 | 0.9 |
| Relative change (%) | -22.0 | -22.0 | -2.9 | 6.4 | 15.9 |

---

## Author Comment (AC2)

**Revision notes for "Attribution of climate change and human activities to streamflow variations with a posterior distribution of hydrological simulations"**

**(hess-2021-528)**

We would like to thank the editors and reviewers for the constructive feedback. We appreciate the valuable and thoughtful comments, which have certainly helped to improve the presentation and quality of our manuscript. We have updated our paper according to your comments and the detailed responses to the comments are described as follows.

Answers to the reviewers in blue.
Modifications of the manuscript in orange.

**To Reviewer #2:**

**General Comments:**

The authors evaluated the attribution of climate change and human activities to streamflow variations with a posterior distribution of hydrological simulations. The contribution distribution has been evaluated in many hydrologic fields using the different methods, however, the posterior distribution has been rarely considered. The author tried to provide a solution to evaluate the attribution of climate change and human activates considering the simulation uncertainty.

**Responses to comments one by one:**

1. Line 65-70, the second typed of method seems to be no difference with the third type of method. For example, I can understand that controlling human activities should be to change the climate factor like the second method simulating multiple scenarios by changing one impact factor. What is the nature difference for the two methods, please describe more clearly.

**Response:** Thanks very much for your comments. Based on your suggestion, we have re-summarized and classified research methods to quantify the contribution rate of climate change and human activities to streamflow change. In the revised manuscript, we re-describe the difference between these three types of methods, and the modified content is as follows:

Line 57-60, Page 2 in the revised manuscript

In general, the commonly used methods of attribution analysis can be divided into the following three categories: 1) conceptual methods, such as the Budyko framework (Li et al., 2007; Liu et al., 2017); 2) hydrological simulation methods (Liu et al., 2019); and 3) analytical methods, such as the climate elasticity method (Liang et al., 2013).

Line 71-74, Page 3 in the revised manuscript

The third type of method is mostly based on numerical calculation, taking the climate elasticity method as an example (Liang et al., 2013), this method introduces the concept of climate elasticity to define the quantitative relationship between changes in streamflow and climatic variables (precipitation, evapotranspiration, etc.), and the CR of HAs to streamflow changes can be obtained by subtracting the CR of climate variables.

**References:**

Li, L. J., Zhang, L., Wang, H., Wang, J., Yang, J. W., Jiang, D. J., Li, J. Y., and Qin, D. Y.: Assessing the impact of climate variability and human activities on streamflow from the Wuding River basin in China, Hydrological Processes: An International Journal, 21, 3485-3491, 10.1002/hyp.6485, 2007.

Liang, K., Liu, C., Liu, X., and Song, X.: Impacts of climate variability and human activity on streamflow decrease in a sediment concentrated region in the Middle Yellow River, Stochastic environmental research and risk assessment, 27, 1741-1749, 2013.

Liu, J., Zhang, Q., Singh, V. P., and Shi, P.: Contribution of multiple climatic variables and human activities to streamflow changes across China, J Hydrol, 545, 145-162, 10.1016/j.jhydrol.2016.12.016, 2017.

Liu, J., Zhou, Z., Yan, Z., Gong, J., Jia, Y., Xu, C.-Y., and Wang, H.: A new approach to separating the impacts of climate change and multiple human activities on water cycle processes based on a distributed hydrological model, J Hydrol, 578, 124096, 10.1016/j.jhydrol.2019.124096, 2019.

2. Line 91-94, as you stated, Farsi and Mahiouri (2019) has also analyzed the uncertainty of hydrological simulations in the process of quantifying the CR of CC and HAs to streamflow changes, however, you thought they only constructed the posterior distribution of the contribution rates of climate change and human activities, and did not specify the accurate contribution rates. I don't think so, Farsi have provided the PDF of contribution rates which can tell high-probability contribution rated clearly. Therefore, what is the innovation for this study differing from the previous studies, not averaging the contribution rates with high-probability NSE.

**Response:** Thank you very much for your comments. We strongly agree with you that Farsi and Mahiouri (2019) have indeed presented a PDF of the contribution rate in their study. However, we believe that our current research has made some expansions on the basis of their research, and the main innovations or expansions are as follows:

1) Farsi and Mahiouri (2019) constructed a PDF of the contribution rate in their study, and compared it with the optimal simulation results of the hydrological model to prove that the results with the highest frequency fully consider the

uncertainty of hydrological simulation. However, the research results presented in Section 5.1 of our study show that the parameter combination with the best simulation performance may have a large difference in the calculated contribution rate from the actual situation.

2) In addition to using the Budyko method to verify the calculation results considering the uncertainty of hydrological simulation, our research also constructed a rough estimation method for quantifying the contribution rate of climate change and human activities to streamflow change in areas where reservoir construction is more active, which can provide method reference for other researchers.

3) According to your suggestion, we added the analysis of parameter uncertainty in the discussion section, and calculated the frame calculation results after excluding unreasonable parameters. This is also an extension of the Farsi and Mahiouri's research to some extent, and it can provide a method reference for the research area with true values of the relevant parameters of the hydrological model.

**References:**

Farsi, N. and Mahjouri, N.: Evaluating the contribution of the climate change and human activities to runoff change under uncertainty, J Hydrol, 574, 872–891, 10.1016/j.jhydrol.2019.04.028, 2019.

3. Line 344-345. The plant-avaiable water coefficient is set to 0.5, why? It is just to match the result of your proposed method? In addition, the contribution method should be not your finding, is it?

**Response:** Thank you very much for your comments. In Zhang's research (Zhang, et al., 2001), it provides the ratio of mean annual evapotranspiration to rainfall as a function of the index of dryness for different value of plant-available water coefficient (Figure 1). For our study area (Lancang River Basin), the multi-year average $E_0/P$ and $ET/P$ values from 1961 to 2015 are 0.55 and 0.96, respectively. Therefore, according to the selection method shown in Figure 1, we set the $\omega$ value to 0.5 in this study.

In the revised manuscript, we have added a description of how the value of $\omega$ is chosen as follows:

Line 354-356, Page 15 in the revised manuscript

According to the method for selecting the value of ω provided in Zhang's research (Zhang et al., 2001), and based on the multi-year average AE/P (0.55) and PET/P (0.96) values in the LR Basin, this study set the value of ω to 0.5.

[Figure]

**Figure 1.** Ratio of mean annual evapotranspiration to rainfall as a function of the index of dryness ($E_0/P$) for different values of plant-available water coefficient ($w$).

Figure 1 How to choose $\omega$-values that apply to different study areas. (Notation: $E_0$ = potential evapotranspiration, ET = actual evapotranspiration)

**References:**

Zhang, L., Dawes, W., and Walker, G.: Response of mean annual evapotranspiration to vegetation changes at catchment scale, Water resources research, 37, 701-708, 10.1029/2000WR900325, 2001.

4. L440, the sensitivity analysis method is not describe clearly, SUFI-2 is a optimal method, and how it conduct parameter sensitivity analysis, what is its relation with Latin hypercube sampling?

**Response:** Thank you very much for your useful comments. As you stated, SUFI-2 is a parameter optimization method and cannot do parameter sensitivity analysis. Therefore, in the revised manuscript, we have revised this part of the content, and the added content is as follows:

Line 456-460, Page 19-20 in the revised manuscript

As described in Section 3.4.2, the sensitivity of 22 selected parameters was evaluated using the SWAT-CUP software (Abbaspour et al., 2007), and this software integrates the global sensitivity analysis method and the parameter optimization methods (such as SUFI-2). The SWAT-CUP can perform a combined optimization and uncertainty analysis using a global search procedure and can deal with a large number of parameters through Latin hypercube sampling.

**References:**

Abbaspour, K. C., Yang, J., Maximov, I., Siber, R., Bogner, K., Mieleitner, J., Zobrist, J., and Srinivasan, R.: Modelling hydrology and water quality in the pre-alpine/alpine Thur watershed using SWAT, J Hydrol, 333, 413-430, 10.1016/j.jhydrol.2006.09.014, 2007.

5. About "equifinality for different parameters" of hydrological simulations. The author selected some experiments with high NSE larger than 0.75 to construct the posterior histogram frequency distribution (PHD) of the contribution rate of climate change and human activities to streamflow changes, and then quantify the contribution rates with higher probability. I think it does not solve the "equifinality for different parameters". The simulations with higher probability still exist the "equifinality for different parameters". To exclude it, the uncertainty (pdf) of parameter should be analyzed to screen out abnormal parameter values. After that, the contribution rates with higher probability are really results excluding the "equifinality for different parameters". These suggestion may be added into the discussion section in revised manuscript.

**Response:** Thanks very much for your comments. We acknowledge that the current methods used in our study have not solved the problem of "equifinality for different parameters" in hydrological simulations, because simulations with higher probability still has unreasonable parameter combinations. Therefore, following your suggestion, in the revised manuscript, we have made the following modifications:

1) In this study, we selected 9 parameters with higher sensitivity among 22 parameters to construct a computational framework to quantify the contribution rate of climate change and human activities to streamflow change. We first analyzed the uncertainty of 9 parameters with higher sensitivity in 575 simulation results (NSE greater than 0.75), and the results are shown in Table 1. From Table 1, we can see that although we have selected a combination of parameters with better simulation performance according to the NSE value (greater than 0.75), these parameters still have great uncertainty (with large 50CI and 90CI values). Among them, the parameter CH_K2, the second sensitive parameter, has the largest uncertainty. For the parameters related to snowmelt runoff (SFTMP and SMTMP), although it has a relatively reasonable median value (-5.39 °C and 0.99 °C, respectively), however, there are still values in the range of 50CI and 90CI that are not in line with their physical meanings. This also means that although the calculation framework proposed in this study can effectively reduce the influence of the uncertainty of hydrological simulation, there are still unreasonable parameter combinations in the calculation process.

Table 1 Uncertainty ranges for 9 highly sensitive parameters

| Parameter | V__ ALPHA_BNK | V__ CH_K2 | V__ SOL_BD | V__ GW_REVAP | V__ SFTMP | R__ CN2 | R__ SOL_K | V__ SMTMP | V__ ALPHA_BF |
|---|---|---|---|---|---|---|---|---|---|
| median | 0.69 | 213.12 | 1.85 | 0.08 | -5.39 | 0.02 | 0.11 | 0.99 | 0.51 |
| 50CI | (0.53,0.85) | (103.6,352.5) | (1.48,2.17) | (0.04,0.13) | (-11.9,1.3) | (-0.09,0.12) | (-0.3,0.48) | (-9.7,11.5) | (0.25,0.75) |
| 95CI | (0.22,0.98) | (14.2,488.0) | (0.96,2.47) | (0.01,0.19) | (-19.4,13.6) | (-0.19,0.19) | (-0.7,0.76) | (-19,19.1) | (0.03,0.97) |

(Notation: 50CI (confidence interval) is expressed by the upper (75%) and lower (25%) bounds of the posterior parameter values among the 575 simulation results; 95CI (confidence interval) is expressed by the upper (97.5%) and lower (2.5%) bounds of the posterior parameter values among the 575 simulation results)

2) Based on your suggestion and the parameter uncertainty analysis in 1).

According to the physical meanings of the 9 parameters with high sensitivity, we selected the values of snowmelt runoff-related parameters (SFTMP and SMTMP) with clear physical meanings as a reference for further research, because we could not obtain the actual values of the other 7 parameters in the Lancang River Basin. After fully collecting the recommendations of relevant references for the value ranges of the two parameters (Abbaspour et al., 2007; Arnold et al., 1998; Yang et al., 2017), this study excluded the parameter combinations outside the recommended value ranges of the two parameters (-5°C≤SFTMP and SMTMP≤5 °C), and finally obtained 55 parameter combinations. According to the selected 55 model simulation results, the contribution rate of climate change and human activities to streamflow change in the Lancang River Basin considering the uncertainty of hydrological simulation was calculated. The results are shown in Figure 2. It can be seen from Figure 2 that among the 55 selected simulation results, 16 calculation results (the most number) indicate that the contribution rate of climate change to the reduction of streamflow in the Lancang River Basin is 45-50% (with an average CR of 47.1%). This calculation result is consistent with the results presented in Fig. 10 which derived from the novel framework proposed in our study. They both show that the contribution rate of human activities to the reduction of streamflow in the Lancang River basin is greater than that of climate change, and the error between the two calculation results is about 4.5%.

[Figure]

Figure 2 Histogram of the number of simulations (exclude parameter combinations with unreasonable values) of the CR (with 5% steps) of climate change to streamflow reduction in the LR Basin at the annual scale and corresponding Nash-Sutcliffe Efficiency box plots.

In general, according to the above research results, it can be seen that the computational framework based on statistical methods used in this study can effectively reduce the impact of uncertainty in hydrological simulations. The reviewer's suggestion in 2) can also provide a method reference for other similar regions, especially for some research areas where hydrological model parameters can be obtained.

Based on the above research results, we have added the following content in the Discussion 5.1 section of the revised manuscript:

In this study, 575 parameter combinations with good simulation results (NSE greater than 0.75) were selected, with a step size of 5%, it is proposed to reduce the influence of hydrological modeling uncertainty on the quantitative results by constructing the posterior histogram distribution of the CR of CC and HAs to watershed streamflow change. However, it is undeniable that there are still unreasonable parameter combinations in the simulation results with high probability (167 times). For the LR basin, it is almost impossible to obtain the measured values of all 9 parameters with high sensitivity (Table 3). Therefore, in order to further explore the possible influence of unreasonable parameter values on the quantitative results, we selected two parameters related to snowmelt streamflow (SMTMP and SFTMP) to exclude unreasonable parameter combinations. According to the parameter value ranges recommended by Abbaspour et al. (2007) and other related references (Arnold et al., 2012a; Yang et al., 2017), in this study, the reasonable value range of these two parameters is set to -5 to 5 ℃. After excluding parameter combinations outside this value range, we obtained 55 simulation results with relatively reasonable parameter values, and the quantization results obtained from this calculation are shown in Fig. 15. It can be seen from Fig. 15 that after excluding unreasonable parameter combinations, the calculated CR of CC in the LR Basin to the reduction of streamflow is 45-50% (with an average CR of 47.1%), and this result is consistent with the results presented in Fig. 10 which derived from the novel framework proposed in our study. At the same time, it is also proved that although the calculation framework proposed in this study may contain unreasonable parameter combinations in obtaining the simulation results with the highest frequency, the calculation results are still highly accurate. In addition, for the research area where the measured values of related parameters can be obtained, the rationality and authenticity of the parameter values should be fully considered while selecting the parameter combination with higher NSE.

**References:**

Abbaspour, K. C., Yang, J., Maximov, I., Siber, R., Bogner, K., Mieleitner, J., Zobrist, J., and Srinivasan, R.: Modelling hydrology and water quality in the pre-alpine/alpine Thur watershed using SWAT, J Hydrol, 333, 413-430, 10.1016/j.jhydrol.2006.09.014, 2007.

Arnold, J. G., Srinivasan, R., Muttiah, R. S., and Williams, J. R.: Large area hydrologic modeling and assessment part I: model development, JAWRA Journal of the American Water Resources Association, 34, 73-89, 10.1111/j.1752-1688.1998.tb05961.x, 1998.

Yang, L., Feng, Q., Yin, Z., Wen, X., Si, J., Li, C., and Deo, R. C.: Identifying separate impacts of climate and land use/cover change on hydrological processes in upper stream of Heihe River, Northwest China, Hydrological Processes, 31, 1100-1112, 10.1002/hyp.11098, 2017.

---

## Referee Report (RR1)

**Review for "Attribution of climate change and human activities to streamflow variations with a posterior distribution of hydrological simulations"**

The authors have addressed all what my concern, especially for the deep analysis on "equifinality for different parameters" of hydrological simulations. however, there are two additional minor revision requiring the authors to answer:

1) The authors have conducted a longer simulation from 1961 to 2015 to analyze the CR of CC and HA; however, first I only see the results from 1961 to 2004, whether the rest results should be described more clearly. In addition, how to deal with land use change in the simulations, which was suggested to be clarified in the revised manuscript. It seems not to be found. As for section 5.2, at least there are four types (1980, 2000, 2010, 2015), how to distribute them to each year's simulation?

2) There may be a mistake. In lines 611-613, it describes the comparison between the impacted period (from 2005 to 2015) and the natural period (from 1961 to 2004); however, in Fig.14, the caption is about from 1970 to 2010.

---

## Author Response (AR2)

**Revision notes for "Attribution of climate change and human activities to streamflow variations with a posterior distribution of hydrological simulations"**

**(hess-2021-528)**

We would like to thank the editors and reviewers for the constructive feedback again. We appreciate the valuable and thoughtful comments, which have certainly helped to improve the presentation and quality of our manuscript. We read the manuscript carefully and answered to two comments made by the Anonymous referee #2. Finally, we also checked the grammar and some expressions in the manuscript in detail, and made revisions throughout the text. All revised have been marked in red.

Answers to the reviewers in blue.
Modifications of the manuscript in orange.

**To Report #1 (Anonymous referee #2):**

**General Comments:**
The authors have addressed all what my concern, especially for the deep analysis on "equifinality for different parameters" of hydrological simulations. however, there are two additional minor revision requiring the authors to answer:

**Responses to comments one by one:**

1. 1) The authors have conducted a longer simulation from 1961 to 2015 to analyze the CR of CC and HA; however, first I only see the results from 1961 to 2004, whether the rest results should be described more clearly. 2) In addition, how to deal with land use change in the simulations, which was suggested to be clarified in the revised manuscript. It seems not to be found. As for section 5.2, at least there are four types (1980, 2000, 2010, 2015), how to distribute them to each year's simulation?

**Response:** Thanks very much for your comments.
1) In fact, in this study, we only calibrated and validated the hydrological model from 1961-2004 (the natural period). As described in Section 3.4.1, we firstly divided the series into natural period and impacted period according to the break point test results of hydrological time series, assuming that the impact of human activities on runoff changes in the natural period is negligible, and then used the natural period data to calibrate and validate the model. Therefore, in other words, we only use the hydrological data during the impacted period to calculate the contribution rate of

climate change and human activities to the runoff change, and did not conduct hydrological simulations during the impacted period.

2) Land use change is often considered to be one of the important factors affecting the hydrological process, therefore, we analyze the land use change process in the LR basin during the historical period in Section 5.2. From the results of Section 5.2, it can be seen that the changes of Permanent glaciers and City are more significant, but they only account for 0.17% and 0.27% of the area of the LR basin, respectively. While the areas of forest and grassland (accounting for 38.4% and 47.2% of the study area, respectively) did not change significantly. Therefore, in the hydrological simulation of this study, the possible impacts of land use change in the study area were not considered. We also provide a supplementary description of the use of LUCC data in the revised manuscript. The added content (Page 6, Line 163 to 164) is as follows:

It should be pointed out that this study only used the land use information in 2010 to construct the SWAT hydrological model, and did not consider the dynamic changes of land use information in the hydrological simulation.

2. There may be a mistake. In lines 611-613, it describes the comparison between the impacted period (from 2005 to 2015) and the natural period (from 1961 to 2004); however, in Fig.14, the caption is about from 1970 to 2010.

**Response:** Thank you very much for your comments. We have corrected this typo in the revised manuscript.

**To Report #2 (Anonymous referee #1):**

Thank you again for all your suggestions and comments on the publication of this paper.